

# Structural complexity and benthic metabolism: resolving the links between carbon cycling and biodiversity in restored seagrass meadows

Theodor Kindeberg[1][*], Karl M. Attard[2,3], Jana Hüller[1], Julia Müller[1], Cintia O. Quintana[2,4], Eduardo Infantes[5]

[1]Department of Biology, Lund University, Sölvegatan 37, 223 62, Lund, Sweden
[2]Department of Biology, University of Southern Denmark, 5230, Odense M, Denmark
[3]Danish Institute for Advanced Study, University of Southern Denmark, 5230, Odense M, Denmark
[4]SDU Climate Cluster, University of Southern Denmark, 5230, Odense M, Denmark
[5]Department of Biological and Environmental Sciences, University of Gothenburg, 451 78, Kristineberg, Sweden

[*]Correspondence: theo.kindeberg@gmail.com

**Abstract.** Due to large losses of seagrass meadows worldwide, restoration is proposed as a key strategy for increasing coastal resilience and recovery. The emergence of a seagrass meadow is anticipated to substantially increase biodiversity and enhance benthic metabolism through increased primary productivity and respiration. Yet, open questions remain regarding the metabolic balance of aging seagrass meadows and the roles benthic communities of the seagrass ecosystem play in overall metabolism.

To address these questions, we investigated a chronosequence of bare sediments, adjacent *Zostera marina* meadows of three and seven years since restoration and a natural meadow located within a high-temperate marine embayment in Gåsö, Sweden. We combined continuous measurements of $O_2$ fluxes using underwater eddy covariance with dissolved inorganic carbon (DIC) and $O_2$ fluxes from benthic chambers during the productive season (July). Based on the ratio between $O_2$ and DIC, we obtained site-specific photosynthetic and respiratory quotients from which we could convert eddy covariance fluxes to DIC. We assessed benthic diversity parameters as potential drivers of metabolic flux variability.

We observed high rates of gross primary productivity (GPP) spanning -18–-82 mmol DIC $m^{-2}$ $d^{-1}$ which increased progressively with meadow age. Community respiration (CR) mirrored the GPP trend, and all meadows were net heterotrophic (GPP < |CR|), ranging from 16–28 mmol DIC $m^{-2}$ $d^{-1}$. While autotrophic biomass did not increase with meadow age, macrophyte diversity did, elucidating potential effects of niche complementarity on community metabolism. These observations provide insights into how community composition and meadow development relate to ecosystem functioning and highlight potential tradeoffs between carbon uptake and biodiversity.



## 1. Introduction

Climate change and concurrent biodiversity loss has motivated restoration of natural ecosystems that can contribute to climate change mitigation, adaptation and at the same time strengthen local biodiversity. One such ecosystem is seagrass meadows, which has suffered substantial losses worldwide during the last century (Mckenzie et al., 2020; Waycott et al., 2009). Due to its foundational role in structuring benthic communities, high productivity and ability to sequester large amounts of carbon, restoring previously lost meadows has been proposed as a low-regret option to address both the climate crisis and the biodiversity crisis concomitantly (Unsworth et al., 2022; Orth et al., 2020; Duarte et al., 2013; Gattuso et al., 2018). Nevertheless, few studies have assessed whether both these goals are mutually attainable within the same restoration projects, or if there are trade-offs between biodiversity conservation and carbon sequestration.

The mechanisms through which a seagrass meadow modifies carbon flows are manifold, influencing both import, export and burial of autochthonous (i.e. seagrass biomass) and allochthonous (i.e. organic matter from other sources) carbon (Duarte and Krause-Jensen, 2017). While sedimentation of allochthonous carbon is largely a passive process ultimately governed by local hydrodynamics, autochthonous carbon sequestration is coupled to the productivity of the seagrass meadow and is thus a function of its metabolic fluxes on timescales ranging from minutes to years (Duarte and Cebrian, 1996; Smith and Hollibaugh, 1993; Smith and Key, 1975). Seagrass community metabolism is comprised of gross primary productivity (GPP) and community respiration (CR) constituted by autotrophic and heterotrophic respiration. The balance between GPP and CR on a daily basis reflects the net metabolism, hereafter termed net community productivity (NCP = GPP - |CR|). The magnitude and direction of GPP, CR and NCP determine all subsequent carbon flows whereby a positive NCP (net autotrophy) equals the net carbon fixed available for remineralization, burial or export (Duarte and Krause-Jensen, 2017). Contrarily, if NCP is negative, the meadow is respiring more organic carbon than is fixed and relies on external or historic inputs of organic matter to sustain metabolism. Empirically assessing community metabolism is thus imperative to constrain a carbon budget and infer the potential net effect of a seagrass meadow on carbon sequestration.

The vast majority of metabolism studies in seagrass ecosystems to date are based on oxygen fluxes (Ward et al., 2022). Converting these fluxes into carbon currency often relies on assuming a constant stoichiometric 1:1 ratio between oxygen and dissolved inorganic carbon ($O_2$:DIC) fluxes which may significantly under- or overestimate actual metabolism (e.g. Turk et al., 2015; Barron et al., 2006; Duarte et al., 2010). For marine sediments, this ratio has been estimated to range between $0.8 - 1.2$ on annual timescales (Glud, 2008 and references therein) but the variability is poorly constrained and likely higher in seagrass systems and on shorter timescales (Trentman et al., 2023; Turk et al., 2015). The discrepancy from a 1:1 ratio between benthic oxygen and DIC fluxes can stem from a wide range of processes, including anaerobic sediment processes, nitrate assimilation, photorespiration and differences in solubility and air-sea gas exchange rates between $O_2$ and $CO_2$ (Weiss, 1970; Trentman et al., 2023). In seagrasses, storage in tissues and transport of oxygen to roots and subsequent radial oxygen loss (ROL) can also contribute to deviations from the theoretical 1:1 relation (Ribaudo et al., 2011; Borum et al., 2007; Berg et al., 2019). Assessing carbon cycling in seagrass meadows without characterizing the marine carbonate chemistry system can thus lead to erroneous conclusions regarding their role in carbon cycling and ultimately their climate change mitigation potential.



Despite the growing number of seagrass restoration projects worldwide, assessments of the effect on
benthic metabolism are lacking. To our knowledge, the only research effort that has specifically addressed benthic
metabolism in restored seagrass was carried out in Virginia Coast Reserve, USA (Rheuban et al., 2014a), where
a large-scale *Zostera marina* restoration project commenced in 2001 (Mcglathery et al., 2012). Rheuban et al.
(2014a) employed a chronosequence approach comprised of a bare site and two stages of development since
restoration (5 years and 11 years) and measured benthic metabolism on diel and seasonal timescales. The authors
found that GPP and |CR| increased up to 25- and 10-fold, respectively with meadow age and this was consistent
through seasons. Yet, NCP was similar, and slightly negative, between the bare site and the oldest restored
meadow on an annual basis, despite the vast differences in autotrophic biomass between the two sites (Rheuban
et al., 2014a). Notably, summer metabolism revealed a net autotrophic state in the five-year-old meadow (NCP),
whereas the older, mature meadow (11yr) had much higher metabolic fluxes and net heterotrophy on the order of
about -50 mmol $O_2$ m$^{-2}$ d$^{-1}$ (Hume et al., 2011; Rheuban et al., 2014a).
Although GPP often substantially increases during summer in temperate seagrass meadows, so does CR
to a similar extent (Ward et al., 2022). Consequently, despite large seasonal variability in photosynthesis and
respiration, the metabolic state (NCP) is often relatively stable on an annual basis, granted there are no major
ecosystem shifts. Interannual variability of NCP has been related to seagrass die-off and recovery episodes (Berger
et al., 2020), and seagrass phenology typically dictate fluxes and metabolic state on intra-annual timescales (e.g.
Champenois and Borges, 2012; Rheuban et al., 2014a). However, a seagrass meadow is comprised of several
components that all contribute to community metabolic fluxes. Aside from the seagrass itself, these include
primary producers such as macro- and microalgae and heterotrophic organisms ranging from macrofauna to
bacteria. Together, these make up the fluxes of $O_2$ and DIC measured in the overlying water column by methods
such as aquatic eddy covariance, benthic chambers or open water mass balance. Isolating fluxes deriving from a
single meadow component is difficult *in situ*, although promising efforts have been made at estimating the role of
benthic fauna in meadow metabolism (Rodil et al., 2021; Rodil et al., 2019a; Rodil et al., 2019b). When planting
seagrass with the stated goals of obtaining both a carbon sink and a biodiversity hotspot, it is essential to
understand the relationship between these two and over what timescales it may change as a meadow develops. It
is therefore necessary to employ a holistic approach and assess biogeochemical and biodiversity parameters in
tandem across multiple stages of seagrass growth. Importantly, both autotrophic and heterotrophic components of
biodiversity are relevant as they are expected to have contrasting effects on metabolism.
The overarching goal of this study was thus to assess how metabolic fluxes and biodiversity change as
the benthic environment evolves from bare sediment to a mature seagrass meadow following active seagrass
restoration. We hypothesized that in an early stage, autotrophic biomass is dominating but total biomass is
relatively low, resulting in small diel variability in metabolic fluxes and a net autotrophic state. As the meadow
grows, fauna colonizes and organic matter builds up, the system goes toward a balanced metabolic state as |CR|
increases relative to GPP. Finally, when the meadow has reached maturity, CR and GPP are tightly coupled in a
system with high turnover and a balanced NCP.
To test these hypotheses, we utilized a chronosequence of four stages of seagrass development since
restoration located within the same sheltered bay. We employed non-invasive, high-resolution aquatic eddy
covariance (EC) in conjunction with benthic chamber incubations (BC). From this we could simultaneously record
fluxes of $O_2$ and carbonate chemistry parameters from which we could evaluate daily metabolic fluxes of both



oxygen and carbon. Additionally, we investigated multiple features of taxonomic and functional diversity of both
macrophytes and benthic fauna and assessed surficial sediment carbon stocks to infer short-term effects of
seagrass restoration on carbon cycling and biodiversity.





## 2. Methods

### 2.1 Site description

The study took place between July 4–20, 2022 in the island of Gåsö (58.2325, 11.3984) located at the mouth of
the Gullmar fjord on the west coast of Sweden (Fig. 1). The bay of Gåsö is a semi-enclosed bay spanning ~0.3
km$^2$ with two narrow inlets and outlets.

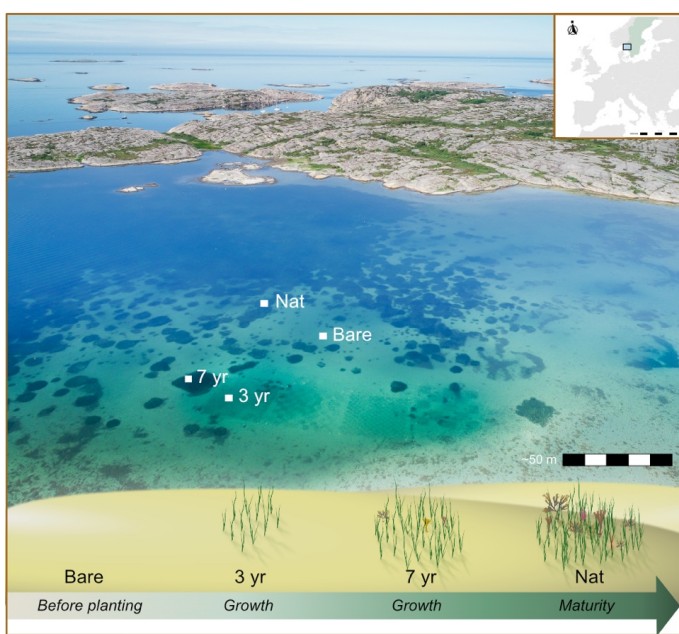

**Figure 1 Aerial view and seagrass development stages after restoration. a) Map showing study location and drone image of Gåsö bay. b) Schematic illustration of seagrass meadow development in the four sites Bare, 3 yr, 7 yr and Nat which represent different stages of meadow development as indicated by the arrow and text in italics.**

The benthos consists of a natural, continuous eelgrass (*Z. marina*) meadow and large patches interspersed with
bare sediment occurring between 1–4 m depth (Fig. 1; Huber et al., 2022). In 2015 and 2019, as part of the seagrass
restoration program ZORRO (www.gu.se/en/research/zorro), two plots of *Z. marina* were planted at the same
depth (~2 m), using the same planting methodology (single-shoot) and shoot density (16 shoots m$^{-2}$) (Gagnon et
al., 2023). These plots thus provided a chronosequence of seagrass meadow ages spanning three and seven years
since planting, while the bare sediment area and the natural meadow corresponded to a 'before' state and a mature
meadow, respectively. The part of the natural meadow we sampled was estimated to have been naturally colonized
by meadow expansion 13–15 years ago (E. Infantes, pers. obs.). Altogether, this yielded four sites within 100 m
distance from each other representing four different stages in the development of a seagrass meadow (Fig. 1).
Importantly, the validity of applying a chronosequence methodology to investigate age-related differences in
seagrass metabolism relies on assumptions that the sites compared experienced similar abiotic conditions after
planting and during sampling. Utilizing adjacent sites within a semi-enclosed bay addresses most of those matters
but to further control assumptions, we monitored *in situ* flow velocity, photosynthetic active radiation (PAR),



temperature, salinity and wind conditions during all deployments and assessed the variation explained by each

variable using linear mixed effects models (see below).

**2.2 Benthic fluxes**

**2.2.1 Aquatic eddy covariance (EC)**

The EC system (Berg et al., 2003) consisted of a stainless-steel tripod frame with an acoustic doppler velocimeter

(ADV Vector, Nortek) and a fast-responding oxygen microsensor (430 UHS, Pyroscience GmbH) programmed

to log data continuously at 16 Hz from co-located measurements of velocity and oxygen concentration. In addition,

two PAR sensors (LI-192, RBR) were mounted to the frame where one was facing upwards to record incident

light and one was directed downwards to record reflected light. This made it possible to calculate the fraction of

absorbed light (*fAPAR*) during deployments. Dissolved oxygen optodes (miniDOT, PME and U26 HOBO, Onset)

were mounted on the leg of the frame and recorded ambient oxygen concentration within the canopy at 1 min

intervals. In addition, a salinity sensor (U24 HOBO, Onset), a turbidity sensor (RBRsolo, RBR) and two light

intensity loggers (HOBO Pendant MX, Onset) were located on the frame recording at 1 min intervals.

**2.2.2 Benthic chambers (BC)**

Benthic incubation chambers consisted of acrylic cylinders (inner diameter = 12.45 cm, length = 65 cm) with a

custom-made motor running a propeller to mix the water within the chamber and avoid build-up of vertical

concentration gradients. We employed pilot tests with dye injection in the laboratory and field to ensure sufficient

mixing and during incubations, chambers were inserted approximately 20 cm into the sediment. We used

transparent (n=3) and opaque (n=3) chambers to simulate day (photosynthesis and respiration) and night

(respiration only), respectively. Upon deployment, chambers were left with lids off for about 30 mins to allow for

suspended sediment to settle.

We drew discrete samples of seawater at onset and termination of each incubation using two 50 mL

syringes attached to 30 cm Tygon® tubing, inserted through a closable sampling port in the chamber lid. We

immediately analyzed seawater in the syringes for pH and dissolved oxygen (DO). pH was measured using an

InLab Micro pH electrode with a FiveGo handheld pH meter (Mettler Toledo). The electrode was calibrated both

using a two-point calibration with standard buffers (pH 7 and 10, Mettler Toledo) at the onset and termination of

the field campaign and calibrated to certified Tris buffer in synthetic seawater (Dr. A. Dickson, SIO) in the

beginning and end of each sampling day. This was done to account for the effect of salinity and to yield values

on the total hydrogen ion scale ($pH_T$). We measured salinity using a conductivity probe connected to a pH/cond

340i multimeter (WTW).

We measured DO using a fiberoptic oxygen sensor coupled to a FireSting® GO2 oxygen meter

(PyroScience). A temperature probe was also connected to the FireSting® to record temperature during each

measurement. Seawater from the syringes was then filtered through 0.45 μm Minisart® sterile syringe filters

(Sartorius) and stored in 50 mL Falcon tubes on ice. Upon return to the laboratory, we placed samples for TA in

a dark container at 4 ºC whereas samples for inorganic nutrients and DOC were frozen (-20 ºC) immediately until

subsequent laboratory analyses.


We determined TA by open-cell potentiometric titration using an 888 Titrando system with an Ecotrode
plus pH electrode (Metrohm). Samples (40–50 g) were titrated with prepared 0.05 M HCl in ~0.6 mol kg$^{-1}$ NaCl,
corresponding to the ionic strength obtained from the mean salinity of the samples. Accuracy and precision (-
1.65±3.76 µmol kg$^{-1}$) were determined using certified reference material (CRM, batch 200, n=8) provided by Dr.
Andrew Dickson at Scripps Institution of Oceanography, San Diego.
We analyzed dissolved inorganic nitrogen (NH$_4$-N and NO$_3$-N) using Flow Injection Analysis on a FIA
Star 5000 analyzer (FOSS) and phosphate (PO$_4$-P) using ion chromatography on an 861 Advanced Compact IC
(Metrohm). We analyzed dissolved organic carbon (DOC) and total nitrogen (TN) using a V-CPH Total Organic
Carbon analyzer (Shimadzu).
We calculated DIC using the package *seacarb* in *R* (Gattuso et al., 2022) with measured values of pH$_T$
and TA in conjunction with in situ temperature, salinity, pressure and NH$_4^+$ as input parameters. We used
dissociation constants K$_1^*$ and K$_2^*$ from Lueker et al. (2000). We also calculated the saturation state of CaCO$_3$
mineral form aragonite ($\Omega_{Ar}$=[Ca$^{2+}$][CO$_3^{2-}$]/K$_{sp}^*$) from each sample using *seacarb*. All solute concentrations were
calculated to µmol kg$^{-1}$, using *in situ* pressure, temperature and salinity data.
Using incubations with discrete measurements to assess flux rates assumes that concentrations change
linearly with time. We verified this assumption *ex situ* by bringing an intact chamber core from the natural
meadow into the laboratory. The chamber was placed in a large water bath with running seawater and prior to
each incubation, the chamber was saturated with oxygen by bubbling compressed air. We ran multiple dark
incubations with continuous logging of dissolved oxygen and temperature (FireSting® GO2) combined with
multiple discrete measurements of pH (n=4) and TA (n=2).

## 2.3 Community components

### 2.3.1 Macrophytes

We evaluated seagrass shoot density by placing a 0.25 m x 0.25 m frame randomly in ten areas of each site and
counting seagrass shoots in subareas of 0.016 m$^2$ (n=10 per site). In addition, we collected seagrass shoots using
a mesh net bag attached to a closable aluminum frame (opening area = 0.1156 m$^2$, n=3 per site). From these
samples, we measured aboveground biomass, shoot density, number of reproductive shoots, leaf length, and
number of leaves per shoot. We also assessed the taxa and biomass of macrophytes other than seagrass (e.g. red
and brown macroalgae). We dried biomass samples at 60 °C for 72 hours and values are reported as dry mass (g
m$^{-2}$).

### 2.3.2 Benthic fauna

We targeted infauna and epifauna separately where we collected epifauna from the mesh net bag samples
described above (mesh size ~ 0.2 mm, n=3 per site). This approach allows for capturing the entire community by
which cores captures infauna and slow-moving epifauna and the mesh net approach captures fast-moving and
larger epifauna. For infauna, we collected sediment cores using polycarbonate cylinders (inner diameter: 7.4 cm,
length: 33 cm, 20 cm depth, n=6 per site) for determination of infauna and seagrass belowground biomass. Upon
return to the laboratory, samples were sieved (0.5 mm) and fixed in 95 % ethanol for subsequent counting and
species identification. Fauna was identified to lowest taxonomic level possible.



### 2.3.3 Sediment properties


In addition to the sediment cores used for infauna, we collected three additional sediment cores from each site to
determine sediment properties. These cores were stored upright and immediately brought back to the lab and
sliced into sections at 2, 4, 6, 8, 12, 16 cm depth. We used the top 0–2 cm section for determination of chlorophyll
*a*, water content, dry bulk density (DBD, g cm$^{-3}$) and porosity (see below). After removing visible root fragments
and shells, we dried remaining sections at 60 ℃ for 72 hours, homogenized with a pestle and mortar and
subsamples (5 mL) analyzed for organic matter content using loss on ignition (4 hours at 520 ℃). Subsamples
from the top 0–2 cm sediment layer (n=12) were also analyzed for particulate organic carbon (POC), particulate
inorganic carbon (PIC) and total nitrogen (TN) using a Vario MAX TN elemental analyzer (Elementar). We pre-
treated samples with HCl to remove carbonates and PIC was obtained by subtracting POC from total carbon (TC).
We obtained a linear relationship between OM and POC (POC=0.47*OM-0.88; R$^2$=0.84, p<0.001) which we used
as a conversion factor to convert remaining OM values to POC and thereby obtain POC values for all core slices.
We calculated carbon density for each slice between 0–12 cm by multiplying POC with surface DBD and
integrated across 0–12 cm to obtain the organic carbon stock (POC$_{stock}$, g m$^{-2}$) in the upper 12 cm of sediment.
Using only DBD values for the top 0–2 cm introduces uncertainty in our depth-integrated POC$_{stock}$ estimates but
a previous study by Dahl et al. (2023) from the same area showed similar DBD values from 0–11 cm (mean
0.43±0.15 g cm$^{-3}$) that were consistent with sediment depth.

### 2.3.4 Chlorophyll a


We collected samples for sediment surface chlorophyll *a* to serve as a proxy for microphytobenthos. From each
sediment core, we used a cutoff 5-mL syringe (Ø=12 mm) to collect 2 mL sediment from the surface layer. This
was repeated three times for each core and we pooled the three samples into one 6 mL sample per core and put in
a 50 mL centrifuge tube covered in aluminum foil to avoid light penetration. The samples were immediately
frozen (-20 ℃) until subsequent extraction and analysis. After thawing at 4 ℃ overnight, we drew a subsample
of 2 mL sediment from each sample, weighed and dried it at 60 ℃ for 72 hours to obtain wet weight (g) dry
weight (g), DBD and water content (%). We extracted the chlorophyll using ethanol (99.5 %) and, after diluting
and incubating overnight, measured fluorescence using a Turner TD-700 fluorometer (Turner Designs). We
calculated chlorophyll *a* content (g m$^{-2}$) using a modified equation from Hannides et al. (2014).

### 2.4 Data analyses


#### 2.4.1 Flux calculations


We calculated oxygen fluxes in the benthic chambers (BC) as the difference in solute concentration between the
onset and termination of each incubation as

$$F_{O_2} \ (\mathrm{mmol} \ O_2 \ m^{-2} h^{-1}) = \frac{\Delta O_2}{\Delta t} \rho h \qquad (1)$$

where $\Delta O_2$ is the change in $O_2$ concentration in mmol kg$^{-1}$ between start and end of incubation, *dt* is the duration
of the incubation in hours, $\rho$ is the density of the seawater in kg m$^{-3}$ and *h* is the height of the chambers from the
top to the sediment surface in meters. We calculated the flux of salinity-normalized TA (nTA = TA/S$_{in\ situ}$ × S$_{mean}$)
in the same way:





$$F_{TA} \text{ (mmol TA } m^{-2}h^{-1}) = \frac{\Delta nTA}{\Delta t} \rho h \qquad (2)$$

Similarly, we used DIC measurements to obtain fluxes as

$$F_{DIC} \text{ (mmol C } m^{-2}h^{-1}) = \frac{\Delta nDIC}{\Delta t} \rho h - 0.5 F_{TA} \qquad (3)$$

where $\Delta nDIC$ is the change in salinity-normalized DIC in mmol kg$^{-1}$. The subtraction of $0.5F_{TA}$ is to account for
the effect of inorganic processes (i.e. calcification/CaCO$_3$ dissolution) on DIC according to the assumptions that
net community calcification affects TA and DIC in a ratio of 2:1 and NCP only modifies DIC (Smith and Key,
1975). $F_{DIC}$ thus represents the DIC flux stemming from primary production and respiration only.
We calculated the photosynthetic (PQ) and respiratory (RQ) quotients from absolute fluxes in transparent
and dark chambers, respectively, as

$$PQ = \frac{|F_{O2,light} - F_{O2,dark}|}{|F_{DIC,light} - F_{DIC,dark}|} \qquad (4)$$

and

$$RQ = \frac{|F_{DIC,dark}|}{|F_{O2,dark}|} \qquad (5)$$

Due to issues with the dark incubations in the natural meadow, RQ from this site was instead calculated as the
average of the three other sites.
We computed EC fluxes from the high frequency time series following a multiple-step protocol described
in Attard et al. (2019). In short, we bin-averaged the time series to 8 Hz, extracted fluxes for consecutive 15 min
time windows using linear detrending (Mcginnis et al., 2014) and corrected fluxes for oxygen storage within the
canopy (Rheuban et al., 2014b). Subsequently, we bin-averaged 15 min fluxes to 1 hr for interpretation. We
defined $F_{light}$ and $F_{dark}$ based on when incident PAR was above or below 1 µmol m$^{-2}$ s$^{-1}$, respectively. All sites
experienced 19 light hours and 5 dark hours on average. We calculated daily metabolic parameters gross primary
productivity (GPP) as

$$GPP \text{ (mmol } m^{-2}d^{-1}) = (F_{light} + |F_{dark}|) \times t_{day} \qquad (6)$$

where $t_{day}$ is the number light hours. We calculated community respiration (CR) as

$$CR \text{(mmol } m^{-2}d^{-1}) = F_{dark} \times 24 \qquad (7)$$

and net community productivity (NCP) as

$$NCP \text{ (mmol } m^{-2}d^{-1}) = GPP - |CR| \qquad (8)$$

We converted oxygen-based daily metabolic fluxes to DIC fluxes by multiplying $F_{light}$ and $F_{dark}$ with our
empirically derived PQ and RQ, respectively:

$$F_{light\_DIC} \text{ (mmol DIC } m^{-2}h^{-1}) = F_{light} \times \frac{1}{\overline{PQ}} \qquad (9)$$


$$F_{dark\_DIC} \text{ (mmol DIC } m^{-2}h^{-1}) = F_{dark} \times -\overline{RQ} \qquad (10)$$

We then recalculated daily metabolic DIC fluxes GPP$_{DIC}$, CR$_{DIC}$ and NCP$_{DIC}$ (mmol DIC m$^{-2}$ d$^{-1}$) using Eq. (6) –
(8). Due to lack of information on the temporal variability in PQ and RQ, we only interpret DIC fluxes on a daily
basis.





### 2.4.2 Biodiversity

We evaluated biodiversity both from a taxonomic and a functional perspective. For taxonomic diversity, we used the *vegan* package in R (Oksanen et al., 2019) to compute Shannon diversity (H') and Pielou's evenness component (J'). H' was converted to effective numbers ($H_{eff}$ = exp(H')) to make it linear and scale to species richness (Jost, 2006). For functional diversity, we first assigned functional traits to each species based on existing literature (Österling and Pihl, 2001; Queirós et al., 2013; Törnroos and Bonsdorff, 2012; Remy et al., 2021; Kindeberg et al., 2022; Riera et al., 2020) and the databases Biological Traits Information Catalogue (Marlin, 2023) and Polytraits (Faulwetter et al., 2014). We then constructed a traits-by-species matrix assigning each species to trait categories (Table S3). To avoid a disproportionally large influence by generalist species on functional diversity, we used fuzzy coding (Chevenet et al., 1994) whereby species comprising multiple trait categories were assigned a score between 0 and 3, with the total sum of each trait always being 3. Based on this matrix, we calculated community-weighted means of trait values (CWM) and several multivariate components of functional diversity including functional richness (FRic), functional evenness (FEve) and Rao's quadratic entropy (RaoQ). These calculations were performed using the *FD* package in R (Laliberté and Legendre, 2010) and further detailed information on these multivariate components and their taxonomic analogs can be found in Mason et al. (2005) and Villéger et al. (2008). As with H', the functional diversity index RaoQ was transformed to effective numbers as $FD_{eff}$ = 1/(1-RaoQ).

We measured biomass divided into classes. We obtained wet weight (g) after blotting each specimen on a tissue for two seconds and dry weight (g) after drying at 60 °C for 24 hours. Regrettably, due to a computer malfunction the class division per sample was lost for infauna samples and only total, pooled biomass per site is available for this group. We combusted pooled samples at 520 °C for 4 hours to obtain ash-free dry weight (AFDW, g m$^{-2}$).

### 2.4.3 Light-use efficiency

We evaluated the relationship between irradiance (PAR) and gross primary productivity (GPP) using a hyperbolic tangent function (Jassby and Platt, 1976; Platt et al., 1980):

$$GPP = P_m \times \tanh\left(\frac{\alpha PAR}{P_m}\right) \qquad (11)$$

where $P_m$ is maximum oxygen flux of gross primary productivity (mmol O$_2$ m$^{-2}$ h$^{-1}$), α is the quasi-linear initial slope of the curve and PAR is seabed irradiance as photosynthetic active radiation (µmol photons m$^{-2}$ s$^{-1}$). We performed curve-fitting in OriginPro 2022 using a Levenberg–Marquardt iteration algorithm, and we scaled the standard error of the fitting parameters with the square root of the reduced chi squared statistic (Attard & Glud 2020).

To examine these relationships further, we calculated the light-use efficiency (LUE) at each site, which indicates the efficiency with which absorbed PAR is converted to primary production, as:

$$LUE = \frac{GPP}{PAR \times fAPAR} \qquad (12)$$

where fAPAR is the fraction of absorbed irradiance calculated from the difference between incident and reflected PAR as measured by the upward and downward facing PAR sensors (see above). Including fAPAR in the calculation of LUE thereby accounts for any differences owing to meadow characteristics such as the higher three-





dimensional meadow complexity (higher fAPAR) relative to bare sediment (lower fAPAR) and captures the diel
differences in seabed reflectance and absorption (Attard and Glud, 2020).

### 2.4.4 Statistical models

To test the effect of differences in abiotic factors between deployments, and thereby validate the use of the
chronosequence approach, we employed linear mixed-effects models (package *lme4* in R (Bates et al., 2015),
testing the effect and interaction of abiotic variables on absolute values of oxygen fluxes ($|F_{O2}|$). We used model
selection (based on Akaike information criterion, AIC) to select the best model, which included sea surface
temperature, flow velocity and PAR as fixed effects and site as a random factor. We used type III ANOVA for
significance testing of fixed effects and likelihood-ratio tests (LRT) for the random effect. Assumptions of models
were tested using the *performance* package in R (Lüdecke et al., 2020). oxygen fluxes calculated by the EC and
BC using a two-sample t-test and compared oxygen and DIC fluxes in the benthic chambers using linear regression
analyses. We tested site differences in biodiversity and sediment parameters using multiple one-way ANOVAs
and visually reviewed multivariate community composition using non-metric multidimensional scaling (NMDS)
and principal components analyses (PCA). We set significance level to α=0.05 for all statistical tests and
performed all analyses in R, version 4.2.3 (Rcoreteam, 2023).

### 2.4.5 Carbon budget

We constructed a carbon budget of daily inorganic carbon fluxes and pools of organic carbon. We based the
sediment carbon pool on the POC stock of the top 12 cm of sediment whereas we inferred seagrass aboveground
and belowground carbon from dry weight (DW) and a global average carbon content for *Z. marina* of 34 % DW
(Duarte, 1990). We estimated macroalgal carbon content based on DW and species-specific carbon content of the
dominant red and brown algae reported from the area, which ranged from 29.1–39.9 % DW (Olsson et al., 2020).
For benthic fauna, we converted ash-free dry weight (AFDW) to carbon, assuming a 50 % carbon content
(Wijsman et al., 1999; Rodil et al., 2021). We converted organic carbon pools to moles and they are reported as
mol C m$^{-2}$. Lastly, we calculated the total pool of organic carbon for each site as the sum of all pool means. We
calculated the total propagated uncertainty (SE$_{total}$) as:

$$SE_{total} = \sqrt{\sigma^2_{sediment} + \sigma^2_{AG} + \sigma^2_{BG}} + \sigma^2_{algae} + \sigma^2_{fauna} \Big/ \sqrt{n} \tag{13}$$

where $\sigma$ is the standard deviation of each pool mean and n is the number of pools. AG and BG is eelgrass
aboveground and belowground biomass, respectively.



### 3. Results

#### 3.1 Environmental conditions

The weather was sunny and dry during all field deployments with only two minor rain events in between (Fig. S1). Sea surface temperature ranged from 17.10–19.98 ºC, driven mainly by the diel light cycle. Salinity ranged from 24.7 to 28.9 but remained constant (±0.1) during each individual deployment. Photosynthetic active radiation (PAR) at the seabed was similar between sites and deployments and reached a highest value of 728 µmol m$^{-2}$ s$^{-1}$ (Fig. 2; Fig. S1). Flow velocities ranged from 0.9 to 21 cm s$^{-1}$, averaging 5.6±3.4 cm s$^{-1}$ across all sites (Fig. 2; Fig. S1).

Ambient seawater chemistry was largely similar between sites, although there was a higher background salinity, TA, DIC and DIN at the 3 yr and Nat site, which were sampled after a weather front passed by likely exchanging some of the bay water with off-shore fjord water (Table S1; Fig. S1). Average DO during deployments was highest in Bare and lowest in 3 yr, averaging (mean±sd) 302.9±21.8 and 260±37.3 µM, respectively.

#### 3.2 Hourly oxygen fluxes

Hourly O$_2$ fluxes followed the diel light cycle and increased both in magnitude and variability going from bare sediments to increasing age of the restored seagrass (Fig. 2).

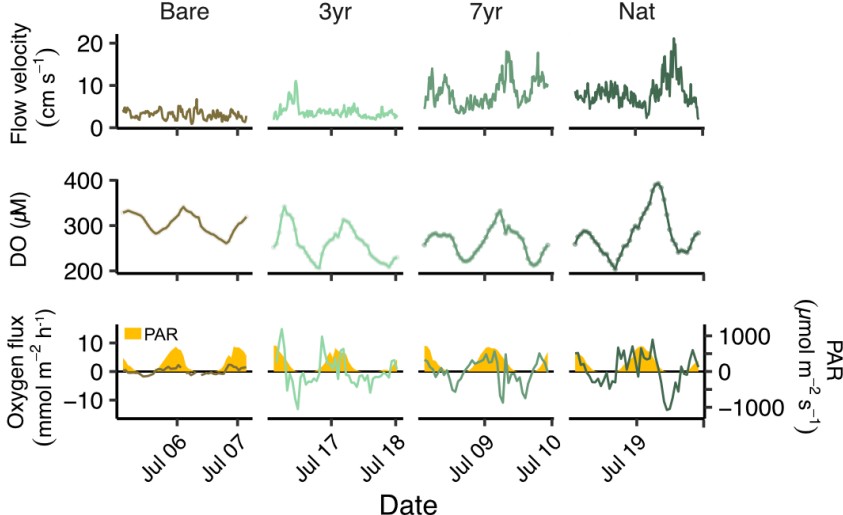

**Figure 2 Time series of flow, oxygen and light. Time series of (a) flow velocity, (b) ambient dissolved oxygen and (c) hourly oxygen flux overlaid on photosynthetic active radiation (PAR) in yellow.**

The largest hourly oxygen fluxes typically occurred in the afternoon, with highest recorded between 14:30–15:30 in the 3 yr site (8.96±1.44 mmol m$^{-2}$ hr$^{-1}$). The largest oxygen uptake rates were generally observed during hours following midnight, with the most negative hourly flux recorded between 23:30–00:30 in Nat (-9.08±5.62 mmol m$^{-2}$ hr$^{-1}$).

Flow velocity was on average significantly higher in the 3 yr compared to Bare and significantly higher in the 7 yr and Nat meadow compared to the 3 yr (Tukey HSD: p<0.05). Although there was a general positive linear relationship between flow velocity and absolute oxygen flux across all deployments, the higher flow



velocities in 7 yr and Nat generally occurred during short time periods at night and did not correspond to consistent
increases in absolute oxygen fluxes for those sites (Fig. S2). Further analysis through linear mixed effects
modelling indicated that while PAR and flow velocity explained a large portion of the variation in hourly $|F_{O2}|$,
the random effect Site was highly significant (LRT = 14.7, p < 0.001) suggesting that some other feature, not
included in the model, contributed to the observed differences in oxygen fluxes between sites.

**3.3 Daily integrated metabolism**

Daily metabolic oxygen fluxes (GPP, CR) as measured by the EC were lowest in the bare sediments and increased
with meadow age (Fig. 3a). GPP and CR were tightly coupled but |CR| was always higher than GPP, amounting
to an average GPP:CR ratio of 0.81 (Fig. 3b). Accordingly, we observed net heterotrophy at all sites (NCP < 0;

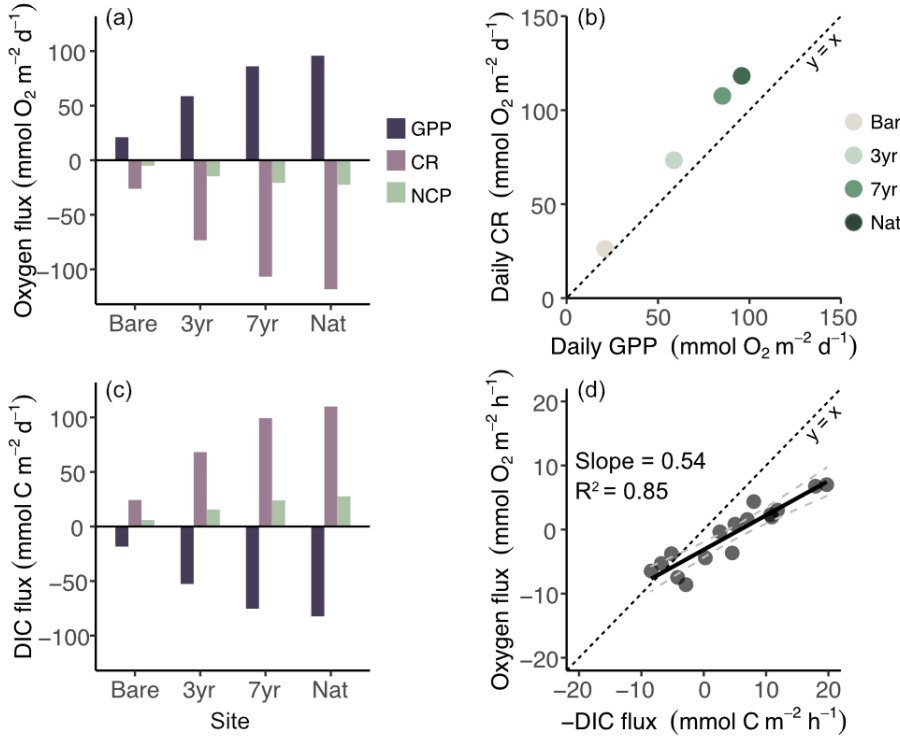

**Figure 3 Fluxes and relationships of oxygen, carbon and productivity dynamics. a) Daily oxygen fluxes of gross primary productivity (GPP), community respiration (CR) and net community productivity (NCP). b) Linear regression of daily oxygen-based GPP and CR; c) Daily dissolved inorganic carbon (DIC) fluxes based on eddy covariance fluxes converted using photosynthetic (PQ) and respiratory (RQ) quotients d) Linear regression between of oxygen and dissolved inorganic carbon (DIC) hourly flux measured in the benthic chambers used to calculate PQ and RQ. Dashed black line indicates slope=1 and dashed grey lines are 95 % confidence interval of the fitted slope.**

Fig. 3a–b). NCP increased three-fold between the bare and the youngest restored meadow (-5 to -15 mmol m$^{-2}$ d$^{-1}$
$^{1}$) with a further 40 percent increase in the seven-year-old meadow (21 mmol m$^{-2}$ d$^{-1}$).

376          Oxygen fluxes measured by the EC and BC were not significantly different from each other (two-sample

t test: p = 0.69), although there was a tendency to overestimate oxygen fluxes in BC relative to EC by 0.7–4.0
mmol m$^{-2}$ hr$^{-1}$. Oxygen and DIC fluxes in the benthic chambers were highly correlated across all incubations (Fig.
3d), irrespective of differences in light conditions. The photosynthetic quotient (PQ) was always less than unity,



averaging 0.46±0.10 across the four sites, whereas the respiratory quotient (RQ) averaged 0.93±0.25. Site-specific
RQ revealed high variability within and between sites ranging from 0.65–1.13.
Estimated DIC fluxes mirrored those of $O_2$ and the benthic DIC net efflux ($NCP_{DIC}$) increased as a
function of meadow age from 6 mmol $m^{-2}$ $d^{-1}$ in the bare sediments to 28 mmol $m^{-2}$ $d^{-1}$ in the natural meadow,
thus confirming the net heterotrophic status of the meadows as determined using oxygen fluxes (Fig. 3b, 3c).
**3.4 Structural and functional diversity**
**3.4.1 Meadow properties**
All three eelgrass sites were characterized by high spatial heterogeneity within each meadow (Table 1). We did
not observe any significant differences in seagrass morphometry such as shoot density, canopy height or biomass.
The only seagrass parameter that differed between the meadows was the number of reproductive shoots containing
seeds that were significantly higher in the natural meadow (p = 0.004). However, the abundance and biomass of
other macrophyte species such as brown and red macroalgae increased markedly with meadow age. In the 3 yr
meadow, only a small specimen of the brown algae *Spermatochnus paradoxus* was found in one sample whereas
in the natural meadow large quantities of up to five different macroalgal species were found. However, due to
large variability in biomass between samples within each site (Table 1), the between-site differences in number
of species were not statistically significant (ANOVA, p>0.05). The composition of macrophyte species became
more even with meadow age such that while the 3 yr meadow was dominated by *Z. marina* (~99 % of total
macrophyte biomass), the 7 yr and the natural meadow had a more heterogenous and evenly distributed
macrophyte community, where *Z. marina* on average contributed 90±15 % and 64±32 %, respectively, to total
macrophyte biomass (Table 1). As a result, the three-dimensional complexity of the canopy increased with
meadow age, driven mainly by large-bodied drifting fucoid species (*F. serratus* and *F. vesiculosus*) and red algae
*Furcellaria lumbricalis* residing, unattached, within the canopies.
Benthic microalgae, as inferred from chlorophyll *a* on the sediment surface, showed the opposite trend
and decreased with meadow age and chlorophyll *a* was significantly lower in sediments underlying 7 yr
(0.28±0.03 g $m^{-2}$) and Nat (0.26±0.01 g $m^{-2}$) compared to Bare (0.56±0.07 g $m^{-2}$).
**Table 1 Eelgrass and macroalgal structural diversity. Morphometrics, biomass and diversity of the sites (mean±SE).**
**'Rep. shoots' represents reproductive shoots with seed spathes present. AG and BG are aboveground and belowground**
**eelgrass biomass, respectively, as captured by sediment cores. Macroalgal biomass represents macroalgae collected in**
**eelgrass canopy samples and maximum number of species refers to the number of macroalgal species found in a sample.**
**Relative proportion is the macroalgal biomass relative to total macrophyte biomass. Species richness, diversity ($H_{eff}$)**
**and evenness (J') refer to macrophytes including macroalgae and eelgrass.**

| Parameter | Unit | 3 yr | 7 yr | Nat |
|---|---|---|---|---|
| **Eelgrass** | | | | |
| Shoot density | $m^{-2}$ | 153±21 | 153±14 | 151±21 |
| Shoot length | cm | 43.3±2.1 | 39.0±0.2 | 40.0±1.2 |
| Rep. shoots | $m^{-2}$ | 9±5 | 3±3 | 32±3 |
| AG biomass | g $m^{-2}$ | 190.4±38.4 | 121.4±17.7 | 151.6±52.0 |
| AG core | g $m^{-2}$ | 117.3±77.8 | 132.5±67.0 | 108.9±49.2 |
| BG core | g $m^{-2}$ | 126.4±63.3 | 259.6±54.7 | 104.3±59.4 |
| AG:BG | - | 2.6±1.6 | 0.5±0.2 | 0.8±0.2 |
| **Macroalgae** | | | | |
| Macroalgal biomass | g $m^{-2}$ | 0.004±0.004 | 16.3±15.4 | 131.6±94.3 |
| Max no. of species | - | 1 | 2 | 4 |
| **Macrophyte diversity** | | | | |




| | | | | |
|---|---|---|---|---|
| Relative proportion | % | 0.002±0.002 | 9.4±8.7 | 35.6±18.4 |
| Species richness | | 1.3±0.3 | 3±0 | 3.3±1.5 |
| Diversity ($H_{eff}$) | - | 1.0±0.0 | 1.3±0.3 | 2.1±0.6 |
| Evenness (J') | - | 0.001±0.0 | 0.2±0.2 | 0.7±0.0 |


### 3.4.2 Benthic fauna

We collected a total of 1927 individuals belonging to 43 taxa. Taxonomic diversity parameters (abundance, number of species, Shannon diversity, evenness) exhibited large within-site variability illustrating the small scale (<10 m) heterogeneity of fauna community structure. These parameters were always higher in vegetated relative to bare sediments but exhibited variable, but generally non-significant, differences between the eelgrass sites (Fig. 4; Table S2). Abundance of infauna was highest in the 3 yr site, dominated by opportunistic polychaetes (e.g. *Capitella capitata*). Yet, the high abundance in the 3 yr site did not result in a corresponding spike in total infaunal biomass but was reflected by the lowest evenness of all sites (J'=0.47±0.08). In the 7 yr, abundance had decreased by a third while species diversity ($H_{eff}$) and evenness (J') nearly doubled, exhibiting similar values as the natural reference meadow (Table S2). Functional trait metrics revealed that both the functional group richness (FGR) and functional diversity ($FD_{eff}$) was significantly higher in 7 yr and Nat compared to 3 yr and Bare which exhibited similar values (Table S2). Functional richness (FRic) was low in the bare sediments (0.06±0.05) and tended to increase with meadow age obtaining highest mean value in the natural meadow (0.53±0.11) but due to high within-site variability, FRic was not statistically different between sites (Fig. 4c).

When separately targeting the meadows for epifauna, we found that they were species rich and highly diverse, ranging from 15–18 species and $H_{eff}$ from 7.4–10.9. However, neither taxonomic nor functional diversity metrics exhibited any significant differences between the meadows, although there were some increasing trends in especially functional evenness (FEve) which was highest in Nat and lowest in 3 yr (Table S2). Epifaunal biomass increased on average three-fold in Nat compared to the two restored meadows and also had the highest within-site variability (15.89±10.48 g m$^{-2}$) driven mainly by gastropods.

Community composition partly shifted as the meadow grew whereby bare sediments and the youngest restored meadow were dominated by polychaetes whereas more epifaunal species such as bivalves and crustaceans increased in older meadows (Fig. 4d). Absolute abundances and biomass supported this, of which also bryozoans and gastropods contributed to higher biomass in Nat relative to Bare. However, multivariate visualization of the different communities indicated that there was much overlap in community composition (Fig. S3).

Based on our functional traits analyses, certain bioturbation modes became more prevalent as a function of meadow age (Table S3). For instance, community-weighted means (CWM) of biodiffusors increased linearly with meadow age and was significantly higher in the natural meadow and 7 yr compared to the 3 yr ($F_{3,20}$ = 8.4; p < 0.001). Surficial modifiers among infauna were higher in eelgrass compared to bare sediment and peaked in the oldest restored meadow at a CWM of 0.29±0.10.





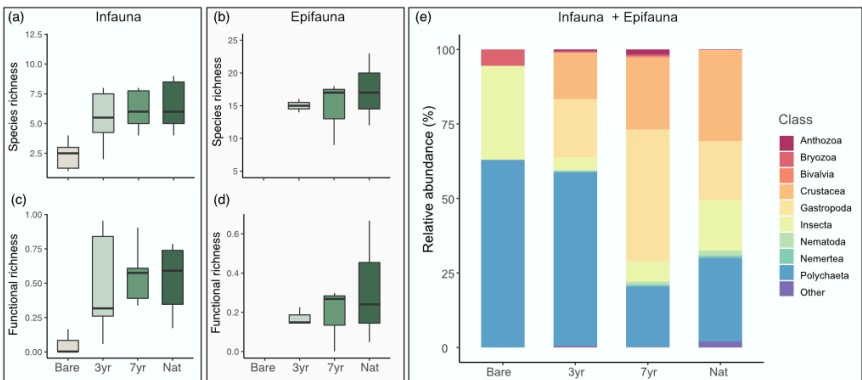


**Figure 4 Biodiversity patterns in benthic fauna. Panels to the left show species richness (a–b) and functional richness (c–d) of infauna samples (a & c) and epifauna samples (b & d). Large panel to the right (e) shows relative abundance of different classes of all fauna combined (infauna + epifauna).**

**3.5 Sediment carbon stocks**
The sediment within Gåsö bay eelgrass meadows has previously been reported as silty sand, with a median grain
size ($D_{50}$) of the surface sediment between 140–170 μm and a silt-clay content of 26-35% (Infantes et al., 2022;
Dr. Martin Dahl, pers. comm.). Concentrations of sediment OM, POC and TN were not significantly different
between sites ($p < 0.05$) and did not display any consistent increases or decreases with meadow age (Table 2).
However, when integrating the POC density across the top 12 cm, the highest POC stock was found in a natural
meadow core (1529 g m$^{-2}$) and the lowest in a bare sediment core (209 g m$^{-2}$). Yet, due to large within-site
variability, the sites were not significantly different from each other ($F_{3,8}$=1.52; p=0.28; Table 2). The lack of a
clear trend with meadow age was further demonstrated by the 3 yr site, which had, on average, 32 % larger carbon
stock than the 7 yr, and the 7 yr site was in turn more similar to the bare sediments site (Table 2). Depth profiles
of POC concentration and density down to 20 cm revealed near-constant values down to between 12–16 cm where
it started to increase (Fig. S4). Natural eelgrass had the highest average POC profile, but average values were
highly skewed by one core replicate displaying POC density four times as high as the other two replicates in the
site.
**Table 2 Sediment properties across sites. Organic matter (OM), particulate organic carbon (POC), particulate**
**inorganic carbon (PIC), total nitrogen (TN) and dry bulk density (DBD) of the top 2 cm of sediment. POC stock is the**
**depth-integrated carbon stock over 0–12 cm sediment depth. Values are mean±SE, n=3 per site.**

| Site | OM | POC | PIC | TN | DBD | POC$_{stock}$ |
|------|------|------|------|------|------|------|
| | (%) | (%) | (%) | (%) | (g cm$^{-3}$) | (g m$^{-2}$) |
| Bare | 3.80±0.23 | 0.88±0.14 | 0.55±0.06 | 0.25±0.02 | 0.37±0.08 | 343±93 |
| 3 yr | 5.42±0.56 | 1.76±0.26 | 0.18±0.04 | 0.34±0.03 | 0.37±0.07 | 652±142 |
| 7 yr | 5.36±0.34 | 1.54±0.31 | 0.61±0.21 | 0.34±0.01 | 0.24±0.03 | 494±39 |
| Nat | 4.98±0.78 | 1.11±0.11 | 0.64±0.30 | 0.29±0.04 | 0.34±0.06 | 883±332 |






### 3.6 Light-use efficiency

All meadows experienced similar incident light conditions (Fig. 5a). The fraction of absorbed light ($f$APAR) was always higher in eelgrass (~97 %) compared to bare sediments (~94 %) but did not differ between eelgrass sites on a daily basis (Fig. 5b). Hourly GPP tracked PAR with a clear hysteresis effect evident in the 7 yr and natural meadow but to a lesser extent in the bare site (Fig. 5c; Fig. S5). P-I relationships were best explained by the hyperbolic tangent function yielding $R^2$ between 0.45–0.74. The irradiance needed for photosynthesis to balance respiration ($I_k$) was almost four times higher in the bare site compared to the 7 yr site, equaling 380 and 97 µmol photons m$^{-2}$ s$^{-1}$, respectively (Table S4; Fig. 5d). Estimated light-use efficiency (LUE) was lowest in Bare (0.001 $O_2$ photon$^{-1}$) and increased with meadow age to 0.004 and 0.005 $O_2$ photon$^{-1}$ in 3 yr and 7 yr, respectively. The highest daily LUE was observed in Nat (0.007 $O_2$ photon$^{-1}$) coincident with the highest number of macrophyte species and the most diverse community structure.

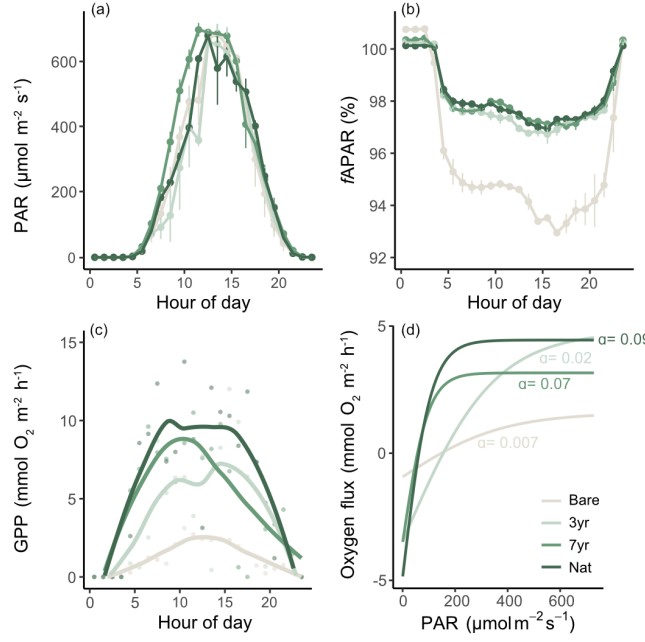

**Figure 5 Light-use efficiency and productivity relationships. Panels a–c show different components of light-use efficiency (LUE) as a function of hour of the day: a) incident photosynthetic active radiation (PAR); b) fraction of absorbed PAR (fAPAR); c) shows gross primary productivity (GPP) as a function of time of day and; d) shows the relationship between oxygen flux and PAR as hyperbolic tangent curves estimated for each site.**

Similar to LUE, GPP and |CR| displayed a positive linear relationship with number of macrophyte species. There was also a positive trend between these parameters and macrophyte Shannon diversity ($H_{eff}$) and the proportion of macroalgal biomass relative to eelgrass biomass, respectively (Fig. 6).

**Table 3 Daily metabolism as a function of meadow age. Curve fitting of daily metabolism parameters GPP, CR and NCP to meadow age (SiteAge) converted to logarithmic scale ($log_{10}(x+1)$). SiteAge for the site Bare was defined as 0 and 13 years for the natural meadow.**

| Metabolic parameter | Function | p | $R^2$ |
|---|---|---|---|
| GPP | $GPP = 67.29 \pm 4.63 \times \log_{10}(SiteAge + 1) - 20.80 \pm 3.65$ | 0.005 | 0.99 |




| | | | |
|---|---|---|---|
| CR | $CR = -83.08 \pm 5.77 \times \log_{10}(\text{SiteAge} + 1) - 26.06 \pm 4.56$ | 0.005 | 0.99 |
| NCP | $NCP = -15.79 \pm 1.26 \times \log_{10}(\text{SiteAge} + 1) - 5.26 \pm 0.99$ | 0.006 | 0.98 |

487

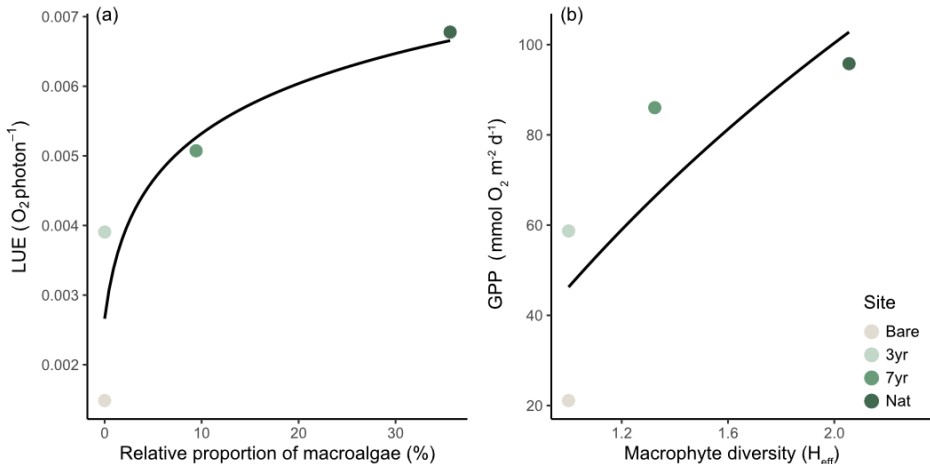

488

**Figure 6 Biodiversity and productivity relationship. (a) Light-use efficiency (LUE) as a function of the relative biomass of macroalgae to eelgrass (R2$_{adj}$= 0.70). (b) Gross primary productivity (GPP) as a function of macrophyte Shannon diversity index (R2$_{adj}$= 0.46); Black lines represent best fit (log$_e$(x+1)). Note that the bare site was not quantitatively sampled for macroalgal proportions or macrophyte diversity and was a posteriori set to 0 % in and 1, respectively, for curve fitting.**

**3.7 Carbon pools**

Converting seagrass community components to carbon illustrates the pools of carbon available for export, remineralization or burial. Notably, total carbon pools were higher in eelgrass relative to bare sediment but were similar between restored and natural seagrass (Table 4). Sediment POC stocks were the largest carbon pools followed by eelgrass biomass which contributed on average 11, 21 and 7 percent to the total carbon pool in the 3 yr, 7 yr and natural meadow, respectively (Table 4).

**Table 4. Carbon pools. Pools of particulate organic carbon (mean±SE, mol m$^{-2}$) in the different components of the benthic habitats. AG and BG are above- and belowground eelgrass biomass, respectively. Fauna is total fauna (infauna+epifauna).**

| Site | Sediment | Eelgrass AG | Eelgrass BG | Macroalgae | Fauna | Total pool |
|---|---|---|---|---|---|---|
| Bare | 28.54±7.71 | 0 | 0 | n.d. | 0.03 | 28.57±13.35 |
| 3yr | 54.28±11.83 | 3.32±2.20 | 3.58±1.79 | 0.00±0.00 | 0.24 | 61.42±10.82 |
| 7yr | 41.15±3.27 | 3.75±1.90 | 7.35±1.55 | 0.45±0.42 | 0.29 | 52.99±4.14 |
| Nat | 73.53±27.65 | 3.08±1.39 | 2.95±0.69 | 3.78±2.76 | 0.56 | 83.91±24.14 |






## 4. Discussion


We present a comprehensive dataset of post-restoration seagrass development that captures several different
components of seagrass metabolism. This enables investigating the role of biodiversity and different components
of a seagrass ecosystem in carbon cycling. We show that i) community-integrated photosynthetic (GPP) and
respiratory (CR) fluxes increase as a function of meadow age (Fig. 3); ii) daily |CR| increased more relative to
GPP resulting in net heterotrophy (NCP<0) on diel timescales during summer; iii) diversity and biomass of
macrophytes other than the restored seagrass could be driving higher primary productivity through increased light-
use efficiency (Fig. 5); iv) faunal communities recover rapidly and attain species- and functional richness
comparable to natural meadows within seven years since restoration (Fig. 4); v) surficial (0–12 cm) sediment
carbon stocks are large but are not significantly affected by the presence of seagrass in this sheltered bay.
Based on the above results, we postulate that while higher diversity of macrophytes contribute to elevated
GPP and CR, the additional CR stemming from benthic fauna communities together with labile organic matter
input push diverse seagrass meadows toward summer net heterotrophy. This illustrates potential tradeoffs between
climate change mitigation and biodiversity conservation as incentives for seagrass restoration. Below we discuss
four main lines of evidence to support this postulation.

### 4.1 Metabolic fluxes scale to meadow development


We found large daily fluxes of GPP- and CR derived $O_2$ and DIC that increased as the system developed from
bare sediments to a mature meadow (Table 3; Fig. 3). Our values are relatively low when comparing to global
average GPP, CR and NCP estimated for temperate seagrasses of 166±14, 130±11 and 34±8 mmol $O_2$ m$^{-2}$ d$^{-1}$,
respectively (Duarte et al., 2010). Yet, it should be noted that discrepancies owing to methodological differences
are difficult to account for. An updated assessment of seagrass NCP in temperate areas reported an average of
29±79 mmol $O_2$ m$^{-2}$ d$^{-1}$, although the study which covered 187 seagrass metabolism studies found that merely 68
% reported net autotrophy (Ward et al., 2022). Accordingly, the notion that seagrass habitats are strongly net
autotrophic is being increasingly contested as methods continue to improve. In all our sampled sites, GPP was
lower than |CR| resulting in net heterotrophy (negative NCP), independently established both by EC oxygen fluxes
and benthic chamber DIC and oxygen fluxes. Several recent studies have reported instances of sustained net
heterotrophy across multiple seagrass species and environments (e.g. Barron et al., 2006; Rheuban et al., 2014a,b;
van Dam et al., 2019; Berger et al., 2020; Attard et al., 2019; Berg et al., 2022, Ward et al., 2022). For instance, a
recent study of *Z. marina* using EC reported GPP and CR values similar to our natural meadow (95 and 94 mmol
$O_2$ m$^{-2}$ d$^{-1}$, respectively) resulting in a near balanced metabolic state across 11 years of monitoring (Berger et al.,
2020). The authors reported a generally balanced metabolic state on monthly timescales but following a
temperature-driven dieback event that diminished seagrass shoot density, GPP and |CR| decreased by 55 % and
48 %, respectively. This shifted the meadow to net heterotrophy during summer (NCP = -26±15 mmol $O_2$ m$^{-2}$ d$^{-1}$
$^{-1}$). In the following years, the gradual increase in seagrass shoot density increased primarily GPP, showing clear
signs of seagrass recovery (Berger et al., 2020).
Although GPP often substantially increases during summer in temperate seagrass meadows, so does CR
to a similar extent (Ward et al., 2022). Consequently, despite large seasonal variability in photosynthesis and
respiration, the metabolic state is often relatively stable on an annual basis, granted there are no major ecosystem



regime shifts. While interannual variability of NCP has been related to seagrass die-off and recovery episodes
(Berger et al., 2020), seagrass phenology linked to abiotic factors such as temperature and light regimes typically
dictates the metabolic state on intra-annual timescales (e.g. Champenois and Borges, 2012; Rheuban et al., 2014a,
b). Here, we show that biotic components other than the seagrass itself can contribute to both the magnitude and
variability in metabolic fluxes. Irrespective of traditional seagrass metrics such as seagrass shoot density and
biomass, GPP and |CR| consistently increased with meadow age which in turn corresponded to increased
autotrophic diversity and macroalgal biomass.

### 4.2 Carbon and oxygen balance

As part of this study, we present a methodological approach that estimates *in situ* DIC fluxes under natural
hydrodynamic and light conditions. This is obtained by combining the advantages of aquatic eddy covariance
with the ability to constrain the marine carbonate system and oxygen dynamics using benthic chambers.
Concurrent deployment of these two methods have been utilized in previous coastal studies (Camillini et al., 2021;
Long et al., 2019; Polsenaere et al., 2021), but only for comparing oxygen fluxes.
Assessing the *in situ* relationship between oxygen ($F_{O2}$) and DIC fluxes ($F_{DIC}$) can provide insights into
biogeochemical processes and renders reliable estimates of photosynthetic (PQ) and respiratory quotients (RQ).
All else equal, photosynthetic and respiratory quotients are governed by the C:N:P ratio of the fixed and respired
organic matter present in the system (Champenois and Borges, 2021). However, considering the various sinks and
sources of organic matter present in a seagrass meadow and the multitude of processes affecting $F_{O2}$ and $F_{DIC}$
differently, this is not very useful. Deviations from the theoretical 1:1 relationship between $F_{O2}$ and $F_{DIC}$ (Fig. 3d)
are thus ubiquitous in the literature (e.g. Turk et al., 2015; Barron et al., 2006; Trentman et al., 2023). In fact, the
slope we observed is identical to what Pinardi et al. (2009) observed in sediments vegetated with the freshwater
macrophyte *Vallisnera spiralis* using sediment cores. Moreover, our relatively low PQ's (0.34–0.52) were similar
to what Ribaudo et al. (2011) observed in *V. spiralis* (0.30–0.68) in microcosms. The authors attributed the low
PQ to oxygen transport to roots and subsequent radial oxygen loss (ROL) which fuels aerobic respiration, a
process well-documented in *Z. marina* as well (e.g. Borum et al., 2007; Jensen et al., 2005; Frederiksen and Glud,
2006; Jovanovic et al., 2015). Turk et al. (2015) observed PQs ranging from 0.5–2.6 in seagrass (*Thallasia*
*testudinium*) and found a temporal component to the variability of PQ with lower values in the morning and higher
in the evening (Turk et al., 2015). Similar to our study, Ouisse et al. (2014) obtained a PQ and RQ of 0.42±0.27
and 0.95±0.22, respectively, using *in situ* benthic chambers in dwarf eelgrass (*Z. noltii*) meadows across several
seasons. The authors hypothesized that the low PQ could also be due to photorespiration in epiphytic algae on the
seagrass leaves which can consume more than three moles of $O_2$ for every mole DIC used (Ouisse et al., 2014).
We observed large quantities of epiphytic microalgae and biofilm on seagrass leaves across all our studied
meadows, albeit only as qualitative observations (Kindeberg, T., *pers. obs.*). However, seagrass epiphytes are
abundant in the area where it can exert detrimental effects on seagrass metabolic performance and positive effects
on epifauna distribution (Brodersen et al., 2015; Gullström et al., 2012; Baden et al., 2010). Other biogeochemical
processes such as production and consumption of highly oxidized photosynthates could be another explanation,
but that is merely speculative at this stage and needs further research. It is important to note that inorganic


processes (i.e. $CaCO_3$ production and dissolution), which can have a large influence on PQ and RQ (Champenois
and Borges, 2021), are implicitly accounted for in our $F_{DIC}$ by subtraction of the $0.5F_{TA}$ term in Eq. (3).

583        While we obtained an average RQ close to unity, it was based on a relatively small sample size compared
to PQ due to issues with dark incubations especially in the natural meadow. It is possible that our acclimation
(~30 mins) or incubation times (~3 hours) were too short for accurately capturing dark DIC fluxes, as seen in the
temporal lag in DIC fluxes relative to $O_2$ fluxes in a study by Fenchel and Glud (2000) and a lag in $O_2$ consumption
due to the primary producer cellular machinery (Tang and Kristensen, 2007). Nonetheless, without any ancillary
data on other biogeochemical processes we cannot reconcile the sources of our observed PQ and RQ.

**4.3 Macrophyte diversity driving light-use efficiency and higher metabolism**
Despite the large research field on the relationship between biodiversity and primary productivity (Tilman et al.,
2014), light-use efficiency (LUE) is largely understudied in benthic metabolism studies (Attard and Glud, 2020).
Studies have hitherto focused mainly on smaller-scale LUE, such as microalgae in microbial mats and corals
(Brodersen et al., 2014; Al-Najjar et al., 2010; Al-Najjar et al., 2012). We observed a positive relationship between
macrophyte diversity and LUE when controlling for biomass, indicating that mixed meadows consisting of both
seagrass and macroalgae utilize light resources more efficiently and are more productive compared to
monospecific meadows. Importantly, the restored seagrass meadows became more mixed over time as drifting
macroalgae inhabited the meadow. These unattached algae are a common feature in the area, often considered a
nuisance that can prevent seagrass recovery (Moksnes et al., 2018). Here it seems they also improve overall LUE
of the meadow and contributes to larger metabolic fluxes.

601        Whether higher LUE is driven by certain species remains unclear, but the change in canopy structure and
increasing three-dimensional complexity can positively influence LUE (Binzer et al., 2006; Zimmerman, 2003).
Niche complementarity is common in ecological systems (Hooper et al., 2005; Loreau and Hector, 2001) and it
is reasonable to believe that with increased diversity of autotrophs, pigment complementarity can facilitate optimal
resource-use, especially as brown and red macroalgae are known to have a range of photosynthetic pigments
(Enríquez et al., 1994). Additionally, the mere presence of multiple growth morphologies may induce self-shading
that further increases LUE (Tait et al., 2014). An increase in photosynthetic pathways (e.g. C3 and C4) with higher
macrophyte diversity and differing affinity for forms of inorganic nutrients (e.g. $NH_4^+$ and $NO_3^-$) is also expected.
Moreover, both *Z. marina* and fucoid species are known to utilize both $CO_2$ and $HCO_3^-$ for photosynthesis (Binzer
et al., 2006). However, the efficiency differs between species (e.g. Larsson and Axelsson, 1999; Invers et al.,
2001) and considering the large spatiotemporal variability in pH and $[HCO_3^-]$ relative to $[CO_2]$ we observed, this
could be another reason for the higher LUE at higher species diversity. Studies from macroalgal canopies have
found similar relationships between macrophyte canopy complexity and LUE, attributed to niche complementarity
where intact assemblages are more efficient and productive than the sum of its parts (Tait and Schiel, 2011; Tait
et al., 2014). For instance, a study by Tait and Schiel (2011) using *ex situ* incubation chambers found that an intact
assemblage of seven species had higher net photosynthesis than the sum of all individual species. The authors
observed that different species played different roles at different irradiances. For instance, the fucoid species
*Cystophora torulosa* was exceptionally efficient at photosynthesizing at high irradiance and did not show signs
of photoinhibition even at PAR > 2000 µmol $m^{-2}$ $s^{-1}$ (Tait and Schiel, 2011). Tait et al. (2014) studied P-I





relationships in macroalgal assemblages and found that when more sub-canopy species where included (up to
four) respiration and photosynthesis increased, thus corroborating our observed trends. However, they found that
production did not saturate at incident irradiance of 2000 µmol m$^2$ s$^{-1}$ as opposed to less speciose assemblages (2
sub-canopy species) that reached light saturation of net primary production (NPP) already at about 600 µmol m$^2$
s$^{-1}$ (Tait et al., 2014). This is somewhat contrary to what we found for GPP, where P-I curves saturated at lower
irradiance with higher macrophyte diversity (Fig. 5d; Fig. S5). Albeit not specifically addressing canopy structure
or diversity, Rheuban et al. (2014a) found that a younger, five-year-old, restored Z. marina meadow was light-
saturated while an older, 11-year-old meadow did not show any signs of light saturation, and this was consistent
across seasons.
Whereas it is rather intuitive that a diverse community of primary producers are better at
photosynthesizing (i.e. higher GPP), the relationship is as strong with CR. This is likely explained by the tight
coupling between GPP and CR stemming from respiration of labile photosynthates (Penhale and Smith, 1977).
However, detritus of macroalgae such as Fucus spp. is also more labile than Z. marina, partly due to a lower C:N
ratio and a more bioavailable polysaccharide composition (e.g. Kristensen, 1994; Thomson et al., 2020).

### 4.4 The role of benthic diversity in seagrass metabolism

The fact that most fauna diversity metrics were not significantly different between the natural meadow and the
youngest (3 yr) meadow implies that benthic diversity recovers quickly. Similar findings have recently been
reported from Z. marina restoration projects in Denmark (Steinfurth et al., 2022) and from the very same sites as
in this study (Gagnon et al., 2023). In fact, Gagnon et al. (2023) found that both taxonomic and functional diversity
recovered within 15 months after restoration but already after 3 months the abundance was similar to documented
abundances in comparable seagrass meadows in the area. The authors partly attributed this to efficient larval
dispersal from the adjacent natural meadow within the bay (Gagnon et al., 2023).
It is generally established that higher diversity yields higher productivity in seagrass meadows (Duffy et
al., 2017), although the mechanisms behind the relationship are debated (Hooper et al., 2005; Gamfeldt et al.,
2015). Based on our results, it seems that high macrophyte and macrofauna diversity positively influence GPP
and CR, respectively, although the relationships with fauna are less clear. Aside from direct cellular respiration,
many infauna species indirectly modify metabolic fluxes across the sediment-water interface through bioturbation
and sediment reworking (Aller and Aller, 1998; Kristensen et al., 2012). A scrutiny of bioturbation and reworking
modes revealed that especially biodiffusors and surficial modifiers increased with meadow age, despite highest
total abundance of infauna in the youngest meadow (Table S2–S3). It is possible that these functional modes
benefited from larger quantities of macroalgal detritus building up on the sediment surface over the years.
Thomson et al. (2020) found the lugworm Arenicola marina, an upward conveyor, contributed to a 37 % higher
efflux of DIC in sediments containing F. vesiculosus compared to Z. marina. Macroalgal detritus was to a much
higher extent respired or consumed compared to seagrass, which instead was buried in anoxic sediment layers by
the lugworm (Thomson et al., 2020). Moreover, the role of bioturbation in oxygenating otherwise anoxic sediment
can have large ramifications for sediment-water fluxes of DIC and could hence contribute to our observed CR.



**4.5 Implications of seagrass restoration on the carbon budget**

The observed net heterotrophy during the productive season implies the system relies on either historic production of autochthonous carbon or on trophic subsidies to sustain metabolism. Albeit only covering a brief period within the summer season, our results suggest that the seagrass in this area receives large amounts of allochthonous carbon that is partly turned over and released as DIC. A large influx and sedimentation of allochthonous carbon was shown in a recent study by Dahl et al. (2023) from the same bay. They reported relatively high carbon accumulation rates ($0.91\pm0.06$ mol m$^{-2}$ yr$^{-1}$) of which 51 % of sediment carbon originated from eelgrass productivity and 38 % from macroalgae (Dahl et al., 2023). Assuming this rate is constant throughout the year ($0.0025$ mol m$^{-2}$ d$^{-1}$), this accumulation rate is about an order of magnitude lower than our measured summer NCP$_{DIC}$, implying that the majority of imported carbon is rapidly remineralized or assimilated by secondary producers (Fig. 7).

Our estimated budget of all carbon pools illustrates that whereas sediment stocks are the dominant pools, organic carbon is built up in living biomass following restoration (Table 4; Fig 7). Eelgrass and macroalgal biomass in the natural meadow made up 58 and 27 % of all biomass, respectively, which is on the same order as the relative proportion of sediment POC sources found in Dahl et al. (2023). Accumulation of sediment carbon and production of living biomass can be decoupled on longer time scales although trophic subsidies (i.e. external inputs) may be required to sustain both (Huang et al., 2015; Cebrian et al., 1997; Duarte et al., 2010).

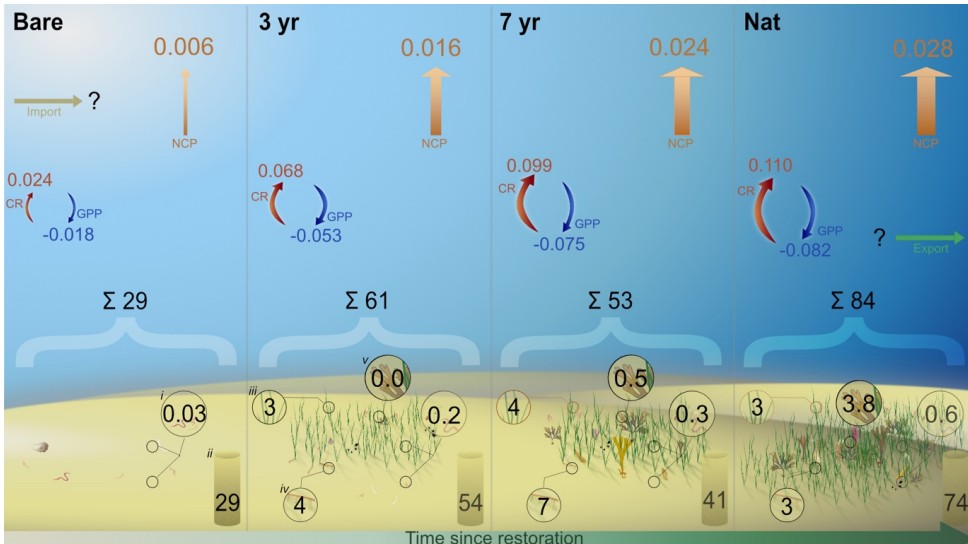

**Figure 7 Pools and fluxes of carbon. Schematic illustration of a carbon budget including different benthic pools (mol C m$^{-2}$) of particulate organic carbon in i) fauna biomass, ii) sediment iii) eelgrass aboveground biomass, iv) eelgrass belowground biomass and v) macroalgal biomass. Arrows indicate the daily metabolic fluxes of dissolved inorganic carbon (mol m$^{-2}$ d$^{-1}$) where blue arrows are gross primary productivity (GPP$_{DIC}$), red arrows are community respiration (CR$_{DIC}$) and orange arrows are net community productivity (NCP$_{DIC}$). Tan and green arrows indicate lateral import and export of particulate organic carbon, respectively.**

While we are able to resolve the dominant carbon pools and metabolic fluxes, the import and export of organic carbon over seasonal timescales is required to reconcile the annual carbon cycling at this site. Nevertheless, it is



reasonable to infer that NCP and carbon sequestration in these seagrass systems are sustained by lateral import of
allochthonous organic carbon.
**5. Conclusion**
Planting seagrass is a profound transformation of the benthic environment that influences biodiversity and carbon
cycling. In this field study, we found that while fauna diversity developed in an expected successional pattern, the
metabolic fluxes and net release of DIC were always higher in seagrass. These fluxes increased with meadow age
and we observed increasing gross primary productivity and respiration as the seagrass grew and drifting algae and
benthic fauna colonized. Collectively, our findings point to a plausible situation where higher macrophyte and
fauna diversity drives high primary productivity and respiration, respectively. Together with ample input of
sestonic organic matter to this sheltered bay, these productive meadows act as effective bioreactors of organic
carbon on diel timescales during summer, as evidenced by the net heterotrophic state and net efflux of DIC. These
results highlight relationships between carbon cycling and biodiversity that should be taken into consideration
when restoring seagrass, especially in sheltered environments with large input of external organic matter. Yet,
identifying the separate mechanisms and constraining the relative importance of fauna and flora diversity for
benthic carbon fluxes remains difficult and should be a focal point in future research.



**Data availability**

The dataset is freely available in the Zenodo repository (https://doi.org/10.5281/zenodo.8363551).

**Author contributions**

TK conceived the study with input from KMA, CQ and EI. TK, KMA, CQ and EI designed the field study. TK, KMA, JH, JM and EI conducted the field work. TK and KMA analyzed data. TK wrote the manuscript with input from all co-authors. All authors approved the submitted version of the article.

**Conflict of interest statement**

The authors declare that the research was conducted in the absence of any financial or commercial relationships that could be construed as potential conflicts of interest.

**Acknowledgements**

TK acknowledges funding from the Gyllenstiernska Krapperup Foundation (grant number KR2020-0066), the European Union LIFE programme (grant number LIFE17 CCA/SE/000048) and the Royal Physiographic Society of Lund (grant number 42518). KMA received funding from the Danish Institute for Advanced Study and CQ received funding from SDU Climate Cluster. We thank Dr. Adam Ulfsbo for assistance with total alkalinity analyses, Dr. Susanne Pihl Baden and Dr. Per Carlsson for help with fauna identification and Dr. Mirjam Victorin for assistance with flux calculations. We are also grateful for the hospitality and assistance of the staff at Kristineberg Center. Symbols used in figures courtesy of the Integration and Application Network, University of Maryland Center for Environmental Science.



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
