# Peer review of "Structural complexity and benthic metabolism: resolving the"

_Biogeosciences, 2023_

## Referee Comment (RC2)

GENERAL COMMENTS:

        The submitted manuscript of Kindeberg et al. under review and discussion for the journal Biogeosciences presents summer benthic community metabolism and composition measurements over a "chrono-sequence" from bare sediments to different restored seagrass meadow development stages in a high temperate marine embayment in Sweden. Different flux techniques (aquatic Eddy Covariance, benthic chamber measurements) along with sediment and benthic fauna/flora characterization and associated computations ($O_2$:DIC ratio, PQ and RQ, LUE, PI curves, etc.) were done to particularly resolve the links between carbon cycling and biodiversity in this restored seagrass meadow area.

The study presents very interesting in situ measurements, analysis and computations through a significant, detailed and well written manuscript. The latter is of particular importance as it identifies mechanisms and links between benthic carbon processes/fluxes and fauna/flora diversity over a restored seagrass meadow system and such coastal carbon research studies need to be done increasingly in the future; thus, I congratulate the authors for their work that is well suited for Biogeosciences journal.

I have two main general comments on the submitted manuscript (i) the first one concerns the lack of certain information especially on benthic flux measurements and computations described in the M&M section that could be given to help the readers to better follow in situ deployments done during the study. ii) my second and major concern is with regards to the chrono-sequence methodology and associated assumptions on which results and discussions relied. Indeed, as authors said, abiotic conditions during the four site measurements have to be not significantly different to truly endorse the four restored seagrass meadow development stage influence only on corresponding measured benthic carbon fluxes. Linear mixed effects models and associated statistical approaches have been used to address this fundamental purpose to validate the approach but associated results are not clear enough or even given in the manuscript as it stands. For instance, flow velocity (i.e. between bare/3years and 7 years/natural sites) as well as salinity and water temperatures during and between this summer week deployments experienced important and significant variations as rightly noticed by authors, that may have influenced benthic metabolism besides meadow habitat development itself. All these aspects need to be better ruled and discussed in the manuscript.

In this way, please see the specific comments below to help in the revision of the different sections and the overall manuscript.

SPECIFIC COMMENTS:

Abstract

- l.32: not clear what are these values? CR? NCP?

- l.33: what about heterotrophic biomass?

1. Introduction

This section is very good as it stands.

2. Material and methods

- l.130: 1-4 m depth, is the studied zone subtidal? what about hydrodynamics, horizontal advection and influence of downstream and upstream systems? Please give general information on it.

- l.136: please give the exact distance between the four sites.

- l.139-143: indeed, these aspects need to be addressed (see general comment above); also, please refer to Table S1 and Fig. S1 here.

- l.147-155: it is very important to refer here to Fig. S1, if not, we don't have any information on EC deployment beginnings and ends (days, dates, numbers, hours, durations at each site), these information have to be given in the text or at least in the Fig. S1 caption. A photo of the EC frame in situ deployed with habitats could be nice in the supplementary material as well.

- l.157-163: similarly, information according to benthic chamber incubations are lacking and must be given in this sub-section: the number of incubations at each site, the order of incubations between clear and dark chambers, the durations of each incubation, the dates of beginning and ending of each incubation, the correspondence with EC deployment (corresponding positions and times?), were they deployed simultaneously with EC measurements? A table with all these EC and BC information could be added in the MS (supplementary material besides the Table S1).

- l.193-194: why $O_2$ concentrations were not measured continuously inside the chamber during each incubation and only at the beginning and at the end of it? With regards to the laboratory experience testing the assumption concentrations change linearly with time, why could authors not test it in situ?

- l.207: the 2.3.4 Chlorophyll a subsection could be displaced right after the 2.3.1 Macrophytes one as we wonder here if microalgae (microphytobenthos) have been measured as well along with macroalgae at each station.

- l.227: how authors are sure the OM versus POC linear relationship they obtained for the top 0-2 cm sediment layer in the 12 samples is the same or is well suited for the other core slices? Is there no variability according to sediment depth for sure?

- l.246: why authors used this flux formulation instead of the one taking into account surface and volume chambers and continuous $O_2$ concentration measurements? (see previous comment above)

- l.249: authors computed salinity normalized TA and DIC fluxes, could they give here the range of salinity they measured at each site during the incubations please?

- l.260: that is why information previously asked in comments above are important to clearly understand what was in situ done in the study.

- l.315-326: Statistical (linear mixed-effects) models used to test the assumptions of similar or non-significant differences in environmental conditions during the 4 stations deployments, measurements are well described here, the presentation of the associated results in the manuscript is another story (see general comment above and other specific comments below). Authors could also better or in a clearer way present in the result section, their statistical tests

and results to show if significant differences in environmental parameters (water temperature, turbidity, current, salinity) existed between each site.

- l.331-332: what about microalgae, was it taken into account here in the carbon budget formulation (sigma algae)?

**3. Results**

- l.342-347: salinity variations from 24.7 to 28.9 are important and could have an (indirect) influence on benthic carbon metabolism at each sampled station. If rain events were minor as author said, could they give explanations on these salinity variations (hydrodynamics?) during this summer week please? Salinity remained constant during each individual deployment, at least between bare and 3 years sites and between 7 years and natural sites according to Fig. S1 (please complete the caption, insufficient information about colors, year, idem Fig. S2), however it is not clear, are there significant variation in salinity values among the four deployments? Please give the same results for all the other abiotic measured parameters (flow velocity, water temperature, turbidity, oxygen concentrations, aquatic PAR, nutrients, TA, DIC etc.) summarized in a table to clearly rule these important considerations according to the author chrono-sequence assumption and possible interference with it in the associated results and discussion. In which sense have flow velocities varied ($0,9-21$ cm s$^{-1}$)? Yet, with these important flow velocity variations, I doubt flow velocity didn't influence at all measured benthic fluxes and did not partly explain associated flux differences observed between sampled stations besides (meadow development) habitats?

- l.348-351: here again, authors have to be clearer, did they measure significant differences in salinity, TA, DIC and DIN between the four sites and between which sites, or not? Please better do the link between these parameter variations and the hydrodynamic of the site. Values given l.351 do not correspond to Table S1 for bare sediment?

- l.362-369: authors clearly computed significant relationships between benthic fluxes and flow velocities, moreover their linear mixed effect models indicated PAR and flow velocity explain a large portion of the variation in $O_2$ fluxes, along with other parameters not included in the model. All these considerations and results should be addressed in a clearer way and discussed after in the discussion section. Is it possible to include in the model parameters such as salinity, temperature for instance?

- l.443 and Fig. 4: only fauna in the sediment (infauna) was encountered at the bare site?

- l.453 and l.459: how authors explain that large within site POC variability?

- l.484 and Table 3: interesting modelling tested by authors, have you tried to test these models with other parameters than the meadow age?

**4. Discussion**

This section is good and well-written. However, with regards to my general and specific comments, it is important to add a sub-section or clear elements on the limitation of the chrono-sequence approach used here due to environmental condition difference observed during this summer week measurement between the four sampled sites.

- l.514-515: please specify here, as the link between both contributions from macrophytes diversity and benthic fauna communities to benthic carbon metabolism is hard to dissociate.

- l.516: it could also be interesting to discuss and compare expected results that could arise at other seasons than summer?

- l.521-522: values could be given here in the discussion as a reminder.

- l.548-549: the sentence is not clear, please specify; if both GPP and CR increased, it does not always imply an autotrophic diversity increase?

- l.578-580: please specify these other biogeochemical processes or delete this sentence since as authors said it is speculative at this stage.

- l.585-588: please specify.

TECHNICAL CORRECTIONS:

- l.44: delete "crisis" and "the" and replace "crisis" by "crise"

- l.126-128 and Fig.1: where are a) and b)

- l.371: the weakest instead of "lowest"

- Fig. 3 caption: delete "of" before oxygen

- l.375: -21 mmol m-2 d-1

- l.452: in "the" natural meadow core instead of "a"

---

## Author Comment (AC1)

**Author response to RC1, Florian Cesbron**

This manuscript is addressing relevant scientific questions within the scope of Biogeoscience focusing on interactions between biological and chemical processes in restored seagrass environments. It combines benthic chamber, aquatic eddy covariance measurements and benthic fauna analysis on a chronosequence of four stages of seagrass development since restoration located within the same sheltered bay. It is well-written, and the methods seem adequate as does the analysis.

We thank Dr. Cesbron for his constructive and detailed review. Please see our response to each of the comments below.

Nevertheless, to better compare your different stages of seagrass development, do you have any complementary information concerning their photosynthesis activity (e.g., PAM fluorescence analysis)?

We did not measure photosynthesis in any other way and have no information on the photosynthesis carried out specifically by the seagrass. This would indeed have been beneficial in discriminating between fluxes due to the seagrass itself as compared to other components of the community.

My technical comments, mostly minor, are provided below:

Is your in-text citations based on relevance? Your preference isn't so clear, sometimes your citations order seems to be chronological but not every time.

We thank the reviewer for catching this error. References have now been updated to chronological order throughout the manuscript.

Methods: Explain more in details your alternative experiment location with your EC/ BC deployments to better understand your operating process (date, distances between EC and BC...) : add more information, for example in a table, about when and where (GPS data) you deployed your EC/BC systems.

We have added a new Table S1 including the location, start, end and duration of EC and BC deployments. In addition, a new Fig. S1 shows a detailed schematic illustrating the positioning of BC's relative to EC together with schematic illustrations of the respective systems. In section 2.2.1 of the Methods, we have added the distances between each deployment location, as measured *a posteriori* using the distance measuring tool in the GIS software QGIS 3.6. We have also added the BC deployments to new Fig. S2 illustrating that BC deployments occurred during EC deployments.

Figure 1: Identify your subdivision a and b on your figure and describe your "Nat" abbreviation as it is described for the first time in a figure.
A figure describing your benthic chamber system could be added or at least a citation using the same system.

We have added a new Figure S1 in the supplementary material which shows a schematic illustration of the EC and BC deployments in addition to detailing the components of the EC and BC.

Modify your "bare" data color coding, e.g., fig3 not the same with your fig 2 and not so visible. Keep your color coding from your fig 2 and modify it even in your supplementary figure.

We thank the reviewer for catching this error and we have updated the color coding throughout the manuscript and supplementary figures.

Line 100: Don't you think you should also add Rodil et al., 2022 (https://doi.org/10.1002/ecy.3648), Rodil et al., 2020 (https://doi.org/10.1007/s10021-019-00427-0)?

We thank the reviewer for these suggestions. We have included Rodil et al., 2022. The reference 'Rodil et al., 2020' was mistakenly named 'Rodil et al., 2019a' in the original manuscript. This reference has now been updated.

Figure 3: delete "of" in your sentence: Linear regression between of oxygen and dissolved inorganic carbon (DIC)"

We have corrected this.

Line 381: A supplementary table showing all detailed PQ and RQ could be added to illustrate your explanation on high-variability.

A specific value for each chamber would not be meaningful since the calculation in Eq. (4) includes fluxes in both light and dark chambers. Calculating chamber-specific PQ and RQ would thus require an arbitrary choice of which dark chamber to subtract from which light chamber. To clarify this, we have added to the section 2.4.1 that we use the average absolute fluxes of light and dark chambers. We also removed the "within-site variability" from section 3.3 which was mistakenly included.

Line 475: Quote your figure 6.

OK, done.

Table 3: You never quote this table in your text except in your discussion (line 522). You could either move it to your supplementary document or delete it.

We have added a sentence referring to table 3 right above the table: "As such, the model that best explained changes in daily benthic metabolism across the four different stages of seagrass development was a logarithmic model (Table 3)."

Table 4: adapt your units as described in your methods part: molC m$^{-2}$.

OK, done.

Line 524: not useful to first compare your data with only one publication without the same methodology. Especially when few sentences later, you are doing a comparison with similar methodology.

We understand the reviewer's point but Duarte et al. 2010 is still a widely used reference for global estimates and we argue that these high values are being increasingly challenged as methods improve (e.g. EC) and more seagrass meadows are sampled. We therefore choose to keep this reference to be able to build that argument in the following text.

Line 572: *Z. noltii* is now called *Z. noltei*

This has now been corrected.

Line 605: a word is missing "to have a large range of photosynthetic pigments".

We have changed to "to have a wide range of photosynthetic pigments"

Figure 7: Lateral import and export need to be documented or delete.

We have removed the arrows for lateral import and export.

Homogenize your reference's part, e.g. (line 767) do not use journal abbreviations.

This has been corrected.

Figure S1: add your site legend on your figure or into your legend. Homogenized your unit's typography: modify wind speed unit m/s by m s$^{-1}$ and flow speed unit cm/s by cm s$^{-1}$.

The figure now has legends and units are corrected. It is now called Fig. S2 after we added a new Fig. S1 describing the methodology (see above).

Figure S2: Add subscripts to your O2 legend in y axis.

Fig. S2 (now Fig. S3) has been updated and subscripts have been corrected.

---

## Author Comment (AC2)

**Author response to RC2, Pierre Polsenaere**

GENERAL COMMENTS:

The submitted manuscript of Kindeberg et al. under review and discussion for the journal Biogeosciences presents summer benthic community metabolism and composition measurements over a "chrono-sequence" from bare sediments to different restored seagrass meadow development stages in a high temperate marine embayment in Sweden. Different flux techniques (aquatic Eddy Covariance, benthic chamber measurements) along with sediment and benthic fauna/flora characterization and associated computations ($O_2$:DIC ratio, PQ and RQ, LUE, PI curves, etc.) were done to particularly resolve the links between carbon cycling and biodiversity in this restored seagrass meadow area.

The study presents very interesting in situ measurements, analysis and computations through a significant, detailed and well written manuscript. The latter is of particular importance as it identifies mechanisms and links between benthic carbon processes/fluxes and fauna/flora diversity over a restored seagrass meadow system and such coastal carbon research studies need to be done increasingly in the future; thus, I congratulate the authors for their work that is well suited for Biogeosciences journal.

I have two main general comments on the submitted manuscript (i) the first one concerns the lack of certain information especially on benthic flux measurements and computations described in the M&M section that could be given to help the readers to better follow in situ deployments done during the study. ii) my second and major concern is with regards to the chrono-sequence methodology and associated assumptions on which results and discussions relied. Indeed, as authors said, abiotic conditions during the four site measurements have to be not significantly different to truly endorse the four restored seagrass meadow development stage influence only on corresponding measured benthic carbon fluxes. Linear mixed effects models and associated statistical approaches have been used to address this fundamental purpose to validate the approach but associated results are not clear enough or even given in the manuscript as it stands. For instance, flow velocity (i.e. between bare/3years and 7 years/natural sites) as well as salinity and water temperatures during and between this summer week deployments experienced important and significant variations as rightly noticed by authors, that may have influenced benthic metabolism besides meadow habitat development itself. All these aspects need to be better ruled and discussed in the manuscript.

In this way, please see the specific comments below to help in the revision of the different sections and the overall manuscript.

We thank Dr. Polsenaere for his time and effort writing this thoughtful and constructive review. We are grateful for the encouraging words and appreciate the discussion on methodological constraints. While the chronosequence allows us to follow contrasting meadows occurring within very close proximity, our ability to measure metabolic fluxes in these different meadows under identical abiotic conditions is challenging and would probably require multiple eddy covariance systems running simultaneously. Unfortunately, this was not possible for this study.

Nevertheless, as rightly pointed out, the chronosequence is based on that general assumption that abiotic conditions that affect metabolism are similar and we agree that it is important to be upfront about this and the topic deserves more attention in the results and discussion.

We do see significant differences between sites with regards to flow velocity, temperature, DO and turbidity. However, these variables together only explain 20% of the variation in absolute oxygen fluxes and the majority of the variation is explained by some other feature(s) not included in the model. We focus the paper on trying to discern what those features may be, and we utilize our light-use-efficiency measurements – which accounts for day-to-day differences to some extent - as a step in our argumentation that it may be partly related to macrophyte structural complexity. While we have added paragraphs in the discussion pertaining to these caveats, we would like to expand on three separate lines of evidence of which we base our inference that any differences in flow velocity between deployments did not consistently explain the observed differences in metabolism between sites.

First, mean flow velocity was indeed different between deployments but the relationship with absolute oxygen fluxes was highly site-specific with generally low $R^2$ values (Bare=0.01; 3yr=0.10; 7yr<0.001; Nat= 0.20) and only in two of the sites (3yr and Nat) was the linear regression slope significantly different from 0. We have updated Fig. S3 (previously S2) to better illustrate this. When relationships are this weak the predictive capabilities are limited, and we would not be able to know what fluxes would have been under different conditions (e.g. lower flow velocity).

Second, our flow velocity measurements were obtained by the ADV mounted on the eddy covariance frame which was positioned within the site/meadow. Regrettably, we did not monitor flow outside of the sites (e.g. incoming flow to the bay). The flow velocities we measured will thus be modulated by the morphology of the bottom substrate (e.g. meadow density, canopy height etc.). Due to differences in drag (bed shear stress) of the different sites and differential modification of hydrodynamics, we cannot discriminate between the actual differences in environmental conditions (due to e.g. weather-induced currents) and what is due to the inherent effect of the bottom morphology on flow. Seagrass meadows are known to have a large impact on flow velocity, and the induced drag will modulate the turbulent flow above the canopy (e.g. Fonseca et al. 1982).

Third, benthic chamber incubations largely preclude any effects of hydrodynamics on fluxes. Nevertheless, we observe a similar increasing trend in FO2 (and decreasing FDIC) in light chambers going from bare to increasing meadow age and oxygen fluxes measured in chambers were not significantly different from eddy fluxes.

Accordingly, it is not possible to unequivocally disprove or validate the chronosequence assumptions based solely on the between-site differences in flow. We have tried to clarify these aspects throughout the results and discussion in the revised version of our manuscript.

Please see our detailed response to each of the comments below.

SPECIFIC COMMENTS:
Abstract
- l.32: not clear what are these values? CR? NCP?

We have added that the values refer to NCP.

- l.33: what about heterotrophic biomass?

This specific sentence regarding niche complementarity refers to macrophytes and we therefore choose to not bring up heterotrophic biomass in this context as to not confuse the readers. We have updated the sentence to clarify this: "While autotrophic biomass did not increase with meadow age, macrophyte diversity did, elucidating potential effects of niche complementarity among macrophytes on community metabolism."

1. Introduction
This section is very good as it stands.

Thank you.

2. Material and methods

- l.130: 1-4 m depth, is the studied zone subtidal? what about hydrodynamics, horizontal advection and influence of downstream and upstream systems? Please give general information on it.

We have clarified that the zone is subtidal and we have added information on tidal amplitude and salinity regimes. Regrettably, additional information on advective flows and hydrodynamics is scarce in the literature.

- l.136: please give the exact distance between the four sites.

We have added the exact distances in section 2.2.1 of the Methods, as measured *a posteriori* using the distance measuring tool in the GIS software QGIS 3.6. Location and duration of each deployment is added to a new Table S1.

- l.139-143: indeed, these aspects need to be addressed (see general comment above); also, please refer to Table S1 and Fig. S1 here.

Please see our response above and detailed responses below. We have added the suggested figure and table reference.

- l.147-155: it is very important to refer here to Fig. S1, if not, we don't have any information on EC deployment beginnings and ends (days, dates, numbers, hours, durations at each site), these information have to be given in the text or at least in the Fig. S1 caption. A photo of the EC frame in situ deployed with habitats could be nice in the supplementary material as well.

We agree and have added the suggested information in a Table S1 and added BC deployments to Fig. S2 (previously Fig. S1). We also have a new Fig. S1 illustrating the EC and BC and their deployment in the field.

- l.157-163: similarly, information according to benthic chamber incubations are lacking and must be given in this sub-section: the number of incubations at each site, the order of incubations between clear and dark chambers, the durations of each incubation, the dates of beginning and ending of each incubation, the correspondence with EC deployment (corresponding positions and times?), were they deployed simultaneously with EC measurements? A table with all these EC and BC information could be added in the MS (supplementary material besides the Table S1).

Please see the new Table S1, Fig. S1 and Fig S2 outlining this information.

- l.193-194: why $O_2$ concentrations were not measured continuously inside the chamber during each incubation and only at the beginning and at the end of it? With regards to the laboratory experience testing the assumption concentrations change linearly with time, why could authors not test it in situ?

We wanted to measure $O_2$, TA and pH (DIC) concurrently from the same sample to get the most robust PQ and RQ. The reason we did not also use continuous loggers was due to logistical constraints and limits of our budget. Since we had six replicate chambers incubating simultaneously in the field we did not have sufficient equipment to continuously monitor DO concentrations in all of them. We therefore used discrete measurements and verified the linear response to time ex situ. We have since repeated incubations in another field study where we have deployed oxygen loggers (miniDOT optodes) within the chambers in situ and those data support the linear concentration changes we observed in the laboratory incubations (unpublished data).

- l.207: the 2.3.4 Chlorophyll a subsection could be displaced right after the 2.3.1 Macrophytes one as we wonder here if microalgae (microphytobenthos) have been measured as well along with macroalgae at each station.

This is a good point and we have moved the chlorophyll a subsection up to subsection 2.3.1 and renamed this "2.3.1 Macrophytes and microphytobenthos"

- l.227: how authors are sure the OM versus POC linear relationship they obtained for the top 0-2 cm sediment layer in the 12 samples is the same or is well suited for the other core slices? Is there no variability according to sediment depth for sure?

Indeed, we cannot know that this relationship is consistent across all core slices. We have added a sentence to clarify this: "This conversion is based on the assumption that the relationship persists with sediment depth and this introduces uncertainty in the POC values at depth."

- l.246: why authors used this flux formulation instead of the one taking into account surface and volume chambers and continuous $O_2$ concentration measurements? (see previous comment above)

Because we have in situ temperature, salinity and pressure for each sample, we can calculate seawater density ($\rho$) using the equation of state, which is included in the seacarb package. We can then obtain gravimetric oxygen concentrations such that:

$O_2$ ($\mu$mol kg$^{-1}$) = $O_2$ ($\mu$mol L$^{-1}$) / $\rho$ (kg L$^{-1}$)

With gravimetric concentrations, using seawater density and height ($\rho$*h) in the flux calculation is equal to using volume divided by area (V/A) when using volumetric concentration units (e.g. mmol L$^{-1}$):

$$\frac{O2\ (volumetric)}{dt} * \frac{Volume}{Area} = \frac{O2\ (gravimetric)}{dt} * \rho * Height$$

*Please see our response regarding continuous vs. discrete measurements of O2 above.*

- l.249: authors computed salinity normalized TA and DIC fluxes, could they give here the range of salinity they measured at each site during the incubations please?

*Salinity was constant at each site during incubations and the range between sites is given in section 3.1. We have added a reference to Table S2 next to the salinity-normalization equation.*

- l.260: that is why information previously asked in comments above are important to clearly understand what was in situ done in the study.

*We agree and hope we have clarified this as per above.*

- l.315-326: Statistical (linear mixed-effects) models used to test the assumptions of similar or non-significant differences in environmental conditions during the 4 stations deployments, measurements are well described here, the presentation of the associated results in the manuscript is another story (see general comment above and other specific comments below). Authors could also better or in a clearer way present in the result section, their statistical tests and results to show if significant differences in environmental parameters (water temperature, turbidity, current, salinity) existed between each site.

*We have added turbidity as a fixed factor in the formulation of the linear mixed effects model (Table S5). We have added turbidity data to Fig. S2 and updated results section 3.1 with a sentence referring to Turbidity results: "Turbidity was generally low but increased markedly at the Nat site, following a minor rain event prior to deployment (Fig. S2; Table S2). Yet, differences in turbidity did not have any detectable effects on seabed PAR (Fig. S2; Table S2)".*

*As described above, salinity could not be included due to missing values. We hope that model results and interpretations are more clearly described now. We have made a new Table S5 with the model output including fixed and random effects.*

- l.331-332: what about microalgae, was it taken into account here in the carbon budget formulation (sigma algae)?

*Microalgae were not considered here. We did not want to extrapolate from chlorophyll $a$ content due to the uncertainties associated with this conversion.*

3. Results

- l.342-347: salinity variations from 24.7 to 28.9 are important and could have an (indirect) influence on benthic carbon metabolism at each sampled station. If rain events were minor as author said, could they give explanations on these salinity variations (hydrodynamics?) during this summer week please? Salinity remained constant during each individual deployment, at least between bare and 3 years sites and between 7 years and natural sites according to Fig. S1 (please complete the caption, insufficient information about colors, year, idem Fig. S2), however it is not clear, are there significant variation in salinity values among the four deployments? Please give the same results for all the other abiotic measured parameters (flow velocity, water temperature, turbidity, oxygen concentrations, aquatic PAR,

nutrients, TA, DIC etc.) summarized in a table to clearly rule these important considerations according to the author chrono-sequence assumption and possible interference with it in the associated results and discussion. In which sense have flow velocities varied (0,9-21 cm s$^{-1}$)? Yet, with these important flow velocity variations, I doubt flow velocity didn't influence at all measured benthic fluxes and did not partly explain associated flux differences observed between sampled stations besides (meadow development) habitats?

Regarding the effect of differing salinity, we agree that e.g. precipitation, submarine groundwater discharge can indeed influence carbon metabolism, and especially total alkalinity (TA). However, comparing our salinity normalized TA flux with non-normalized TA flux equals a mean offset of only 0.7±1.9 mmol m$^{-2}$ hr$^{-1}$. Furthermore, we do not observe any relationship between salinity and O$_2$ or DIC fluxes across our benthic chamber incubations. To clarify how abiotic conditions varied, we have made a new Table S2 which includes the mean±sd of PAR, temperature, flow velocity, DO, salinity and turbidity measured during EC deployments. We have also updated the captions to Figure S1 and S2 (now Fig. S2 and S3) and added the incubations to Fig. S2. Regarding nutrients, TA and DIC we only have data on these from the incubations, and the ambient concentrations at the onset of each incubation is listed in Table S3.

Flow velocity did indeed correlate positively with oxygen flux across all sites. However, flow velocity explained 20% or less of the variance, and there was no linear relationship in Bare (R2=0.01; p=0.22) and 7 yr (R2<0.001; p=0.98). Because of the contrasting relationships with oxygen flux we cannot constrain the effect it may have had on between-site differences in metabolism.

- l.348-351: here again, authors have to be clearer, did they measure significant differences in salinity, TA, DIC and DIN between the four sites and between which sites, or not? Please better do the link between these parameter variations and the hydrodynamic of the site. Values given l.351 do not correspond to Table S1 for bare sediment?

DO values on l.351 correspond to mean ambient DO as measured during EC deployments whereas Table S3 (previously Table S1) are starting conditions of the incubations only (which lasted 3 hours out of the 48 hours of EC deployment). We have clarified this in the sentence and refer to the new Table S2, which includes mean±sd of abiotic conditions including post hoc tests of differences between sites from one-way ANOVAs.

- l.362-369: authors clearly computed significant relationships between benthic fluxes and flow velocities, moreover their linear mixed effect models indicated PAR and flow velocity explain a large portion of the variation in O$_2$ fluxes, along with other parameters not included in the model. All these considerations and results should be addressed in a clearer way and discussed after in the discussion section. Is it possible to include in the model parameters such as salinity, temperature for instance?

We agree with the suggestions and realize that the manuscript would benefit from increased clarity on these aspects.

Indeed, there is a significant positive relationship between flow and abs(O2_flux) across the entire study, but site-specific relationships range from non-existent to significantly positive (see above). We anticipate PAR to explain a large part of daytime O2 fluxes in autotrophic environments as a primary driver of photosynthesis and this we discuss in relation to our P-I

curves. Importantly, there were no between-site differences in neither daily integrated PAR nor the mean PAR during deployments (Fig. 5, S2). Regarding the phrasing that flow and PAR explain a large proportion of the variation in O2 fluxes this is in relation to the other abiotic variables in the model (i.e. Temp and now also turbidity). It still does not explain a lot of the total variation in O2 fluxes, only 20% as mentioned above. We updated the text to clarify this:

"Consequently, site $R^2$ values were low ranging from <0.001 – 0.20 (Fig. S3). Further analysis through linear mixed effects modelling indicated that while temperature, PAR, turbidity and flow velocity explained 20% of the variation in hourly $|F_{O2}|$ across all sites, the random effect Site was highly significant (LRT = 20.9, p < 0.001) suggesting that some other feature, not included in the model, contributed to the observed differences in oxygen fluxes between sites (Table S5)."

Regarding the last part, we have updated the model to also include turbidity and created a new table S5 with the model formulation and its results. As for salinity, we unfortunately do not have continuous salinity measurements at all sites during EC deployments due to a logger malfunction and we can therefore not include it in the model of hourly O2 fluxes. Nevertheless, salinity was highest in Nat and 3 yr and lowest in 7yr and Bare which is not consistent with any trend in daily metabolism for those sites.

- l.443 and Fig. 4: only fauna in the sediment (infauna) was encountered at the bare site?

Epifauna (using the mesh net frames) was only sampled in the seagrass. We have clarified this in the methods section (2.3.2) and added "n.d." to the bare site in Fig. 4 b & d.

- l.453 and l.459: how authors explain that large within site POC variability?

We could only speculate but such large small-scale spatial variability is not uncommon for POC profiles in eelgrass meadows in the area (e.g. Röhr et al. 2016; Dahl et al. 2020).

- l.484 and Table 3: interesting modelling tested by authors, have you tried to test these models with other parameters than the meadow age?

Yes, and similar relationships emerged with e.g. macrophyte diversity (as illustrated in Fig. 6). Here we focus on meadow age as we believe it corresponds best to our main hypothesis and aim of the study.

4. Discussion

This section is good and well-written. However, with regards to my general and specific comments, it is important to add a sub-section or clear elements on the limitation of the chrono-sequence approach used here due to environmental condition difference observed during this summer week measurement between the four sampled sites.

We agree with the lack of clarity on these topics and have added a paragraph discussing the limitations of the chronosequence approach with respect to abiotic conditions:

"The chronosequence approach employed in this study utilizes the unique opportunity of assessing contrasting restored seagrass habitats of different ages that exist within a close

distance from each other (Fig. 1). This enables comparisons between near-identical geomorphology, bathymetry, hydrodynamics and seawater characteristics. However, due to logistical limitations we were unable to measure all four sites simultaneously leading to a temporal mismatch of these comparisons. Consequently, this introduces the risk of potential environmental changes between deployments. Importantly, if the change in environmental conditions is conducive to altered benthic metabolism it can influence the comparison along the chronosequence (i.e between sites). The combined effect of abiotic variables, including PAR, flow velocity, seawater temperature and turbidity accounted for 20 % of the variation in $O_2$ fluxes, as measured by the eddy covariance. Noticeably, PAR reaching the seabed did not differ between sites, despite varying levels of turbidity (Fig. 2; Fig. S2). Salinity was higher in the 3 yr and Nat site compared to 7 yr and Bare (Table S2; Fig. S2). However, due to missing data, we could not evaluate its impacts on oxygen fluxes within the model. However, we found no discernable effects on either oxygen or carbon fluxes during our incubations, suggesting that variability in salinity was not a driving factor of metabolism. Flow velocity peaked in Nat and 7 yr sites but while there was a positive relationship between $|F_{O2}|$ and flow in Nat and 3 yr site, no such relationship was evident in the 7yr or Bare site (Fig. S3). Nonetheless, we cannot decisively rule out the potential role of varying flow velocities in the observed differences in benthic metabolism between sites."

- l.514-515: please specify here, as the link between both contributions from macrophytes diversity and benthic fauna communities to benthic carbon metabolism is hard to dissociate.

This paragraph is merely an opening paragraph to the following discussion and the roles of macrophyte and fauna diversity in carbon metabolism are specifically addressed in section 4.3 and 4.4, respectively.

- l.516: it could also be interesting to discuss and compare expected results that could arise at other seasons than summer?

Yes indeed. We have added a sentence as the end of section 4.1 to this effect: "Further research should address whether these relationships are consistent across seasons and what role differing macrophyte phenologies play."

- l.521-522: values could be given here in the discussion as a reminder.

We agree and have added the average GPP, CR and NCP of the three seagrass sites.

- l.548-549: the sentence is not clear, please specify; if both GPP and CR increased, it does not always imply an autotrophic diversity increase?

Yes, since we are discussing the absolute value (|CR|) the correlation is indeed positive for both. We have updated the text to clarify this: "Irrespective of traditional seagrass metrics such as seagrass shoot density and biomass, GPP and |CR| consistently increased in magnitude with meadow age which in turn corresponded to higher autotrophic diversity and macroalgal biomass."

- l.578-580: please specify these other biogeochemical processes or delete this sentence since as authors said it is speculative at this stage.

We have removed this sentence.

- l.585-588: please specify.

We have specified the average (±sd) incubation time of 3.0±0.1 hours but unfortunately, we do not have exact data on acclimation times for each chamber. We started incubations 30 minutes after deployment and the order of chambers was random to avoid making acclimation times a potential systematic error.

TECHNICAL CORRECTIONS:
- l.44: delete "crisis" and "the" and replace "crisis" by "crise"

- l.126-128 and Fig.1: where are a) and b) - l.371: the weakest instead of "lowest"
- Fig. 3 caption: delete "of" before oxygen - l.375: -21 mmol m-2 d-1

- l.452: in "the" natural meadow core instead of "a"

We thank the reviewer for catching these typographical errors and we have corrected them accordingly, except the last comment (l.452) in which the value refers to one core out of three. Here we have changed to "one".

**References:**

M. Dahl, M. E. Asplund, D. Deyanova, J. N. Franco, A. Koliji, E. Infantes, et al., 2020. High seasonal variability in sediment carbon stocks of cold-temperate seagrass meadows. Journal of Geophysical Research: Biogeosciences, 125:e2019JG005430 https://doi.org/10.1029/2019JG005430

M. S. Fonseca, J. S. Fisher, J. C. Zieman and G. W. Thayer, 1982. Influence of the seagrass, Zoster marina L, on current flow. Estuarine Coastal and Shelf Science 1982 Vol. 15 Issue 4 Pages 351

Röhr, M. E., Boström, C., Canal-Vergés, P., and Holmer, M. 2016. Blue carbon stocks in Baltic Sea eelgrass (*Zostera marina*) meadows, Biogeosciences, 13, 6139–6153, https://doi.org/10.5194/bg-13-6139-2016

---

## Author Comment (AC3)

**Author response to RC3, Guillaume Bernard**

The article authored by Kindeberg *et al.* presents the results of a study aiming at exploring the role of seagrass meadows and associated biodiversity for carbon cycling at different stages of restoration process. The article, through the deployment of various techniques allowing for the measurement of oxygen and dissolved carbon fluxes as well as associated sediment and biodiversity components characteristics during the productive season, provides very interesting, and potentially important, results for a better understanding of seagrass ecosystems metabolism and functional effects of restoration actions. The article is well written and data analyses as well as subsequent interpretations are appropriate and convincing. I therefore only made a few comments/questions for the authors, listed below:

We thank Dr. Bernard for his helpful review. We are especially grateful for the reviewer pointing out that lack of information on fauna traits, and we have tried to amend this in the revised version. Please see our detailed response to each of the comments below.

-l157-178: Please give more information about the temporal sampling strategy. This temporal deployment strategy can indeed be deduced for EC measurements in figure 2, but this must be presented and justified already in the Methods section. Along the same line, have the BC deployments been carried out simultaneously (or kind of) with EC ones?

We have added a new Table S1 that outlines the timing and duration of EC and BC deployments. We have also added shaded areas to an updated Fig. S2 (previously S1) that indicates when BC incubations occurred. Indeed, incubations were performed during EC deployments.

L213: Please specify what does exactly below ground biomass mean. All roots and rhizomes? Only living ones?

We have added the following sentence to section 2.3.1: "Seagrass belowground biomass including live and dead roots and rhizomes were collected using sediment cores (see below)".

L220: In line with my just above comment, have the visible root fragments also been removed for the POC measurements? It must be specified because it can have implications for the carbon pools calculations in the case where only living roots and rhizomes would have been quantified as below ground biomass.

Yes, indeed. We have clarified that we removed all visible roots and rhizomes and rephrased the sentence so that it is clear that all core slices including those used for POC were included.

L256: please precise what does "absolute fluxes" mean. Averages of the three replicates?

Because photosynthesis-derived DIC and O2 fluxes have opposite signs (following the equation of photosynthesis) we have used the absolute values. We have updated the sentence to clarify what is meant: "We calculated the photosynthetic (PQ) and respiratory (RQ) quotients from the average absolute fluxes (i.e. the magnitude of the flux, excluding the direction) in transparent and dark chambers, respectively, as […]"

L281: please introduce in the text the traits that have been chosen, and potentially the rationale for these choices relative to the research question. Were they all explicitly linked to carbon and organic matter cycling?

We have added the following sentence: "The selection of functional traits was based on direct connections to carbon cycling including feeding mode, bioturbation mode and whether the species is calcifying. Indirect, general traits included movement mode, living habit and environmental position. This selection process resulted in 25 trait modalities from which we constructed a traits-by-species matrix assigning each species to specific trait modalities (refer to Table S4). Species can exhibit multiple trait modalities, depending on life history and environmental conditions. To address this, and to avoid a disproportionally large influence by generalist species on functional diversity, we used fuzzy coding (Chevenet et al., 1994) whereby species comprising multiple trait modalities were assigned a score between 0 (no association) and 3 (full association), with the total sum of each trait always being 3.."

We have also updated Table S4 to reflect this and updated the fuzzy coding for Calcareous/Non-calcareous to also include crustaceans with low $CaCO_3$ that were previously assigned 0. We changed the values of FRic and FEve in Table S6 accordingly but these minor changes did not affect any conclusions.

Figure 2 and lines 353-384: Figure 2 shows that flow velocities varied across studied patches. What are the implications of these differences for the surfaces (areas) of the footprint from which fluxes are integrated? Were these footprint areas comparable across patches? How did they coincide with the surface of the studied patches themselves?

In theory, the size and shape of the EC footprint is independent of the horizontal flow velocity, because in the model by Berg et al. (2007), the horizontal velocity scales with the vertical. We have added discussion on flow velocity variability, also in response to comments put forward by Reviewer 2.

L387-388: It is not clear whether this absence of significance also apply for below-ground biomass or not. It is important to precise this because, as shown in the table 2, BG core is clearly higher in the 7 yr meadow compared to either 3 yr and Nat. Along the same line, please indicate in the table the significant differences when there are, to help the reader.

According to one-way ANOVA, BG core is not significantly different between sites ($F_{2,15}$=2.79; p=0.09) although the site mean of 7 yr indeed is higher. The within-site variability is so large that it exceeds the between-site variability. We have added an asterisk to indicate significance in Table 2 showing that only the number of reproductive shoots exhibited a statistically significant difference between sites (p<0.05).

L438: bioturbation modes must be presented earlier, i.e. in the method section, see my comment above about traits.

We have clarified the functional traits in the Methods section (see our comment above) and specified the different modalities in the Table S4: "Feeding modes are suspension feeder (SuspFed), surface detritivore (SurfDet), burrowing detritivore (BurrDet), Predator (Pred), grazer/herbivore (GrazHerb) and omnivore (Omni). Bioturbation modes are biodiffusors (Biodiff), upward conveors (Upconv), downward conveyors (Downconv), surficial modifier (Surfmod), Regenerator (Reg) or not relevant (NotRel). Living habits are free living (Free), burrow dwelling (BurrDwell), tube dwelling (TubeDwell) or attached (Attach)."

**References**

Berg, P., Røy, H., & Wiberg, P. L. (2007). Eddy correlation flux measurements: The sediment surface area that contributes to the flux. *Limnology and Oceanography*, *52*(4), 1672-1684.